# MULTIVARIATE TIME SERIES IMPUTATION WITH SIGNAL–NOISE DISENTANGLED GRAPH PROPAGATION

## ABSTRACT

Missing data are pervasive in real-world multivariate time series, particularly in large-scale, high-frequency systems. Although recent graph-based and transformer-based methods achieve state-of-the-art (SOTA) performance by performing spatial graph propagation or leveraging self-attention mechanisms, they suffer from two key limitations: (1) treating each time series as an indivisible whole, without uncovering its internal temporal dynamics, and (2) relying on linear projections to connect spatial and temporal representations, which insufficiently depicts the complex spatial-temporal interactions. Motivated by the above limitations, we propose **GraphTSI**, a **Graph**-based multivariate **T**ime **S**eries **I**mputation method with signal-noise decomposition, where the signal component captures predictable dynamics and the noise component reflects unpredictable exogenous shocks of time series. To enable robust decomposition, we propose a prediction–subtraction framework where the prediction step progressively estimates predictable signal component, while the subtraction step uses the discrepancy between this estimate and the observed values to extract the exogenous noise component. Furthermore, for effective spatial-temporal interactions, we build an augmented bipartite graph that captures adaptive, non-linear transformation between spatial and temporal dimensions, and propagates signal and noise components through neighboring time series. Extensive experiments across nine datasets from three real-world domains demonstrate the superiority of GraphTSI, with average MAE improvements of 10.273% and 17.580% over graph-based and transformer-based SOTA methods, respectively.

## 1 INTRODUCTION

Multivariate time series data are among the most widespread data types in real-world applications, such as air quality monitoring (Yang et al., 2025a; Hoinaski et al., 2025), energy production or consumption regulating (Anonto et al., 2025; Jafarigorzin et al., 2025; Meha et al., 2025), and traffic analysis (Li et al., 2025; Gong et al., 2025). However, real-world time series are often incomplete due to equipment malfunctions, communication errors, or other data collection errors (Chen & Sun, 2021). The missing data prohibit downstream tasks like prediction (Wu et al., 2021; Zhou et al., 2022), classification (Wen et al., 2025), and data-driven analysis (Han et al., 2025; Wang et al., 2024) as they generally assume complete time series as inputs. To address this issue, multivariate time series imputation (Wang et al., 2025b) is studied to estimate missing values from observed data, ensuring the completeness of the time series for subsequent downstream applications.

In the literature, many works have been proposed for time series imputation. Early methods, such as weighted averaging and statistical models (Bashir & Wei, 2018; Moritz & Bartz-Beielstein, 2017), are used for imputation but fail to capture nonlinear patterns or long-term dependencies. Matrix factorization (LIU et al., 2023b; Yu et al., 2016; Chen et al., 2022) is employed to learn to fit on observed values with low-rank matrix multiplications to impute missing values, but their designs focus on in-sample fitting and lack temporal predictive capability. Generative methods (Yang et al., 2024; Islam et al., 2025) are also widely used to capture the data distribution for imputation, but suffer from training instability. Modern recurrent neural network (RNN)-based models (Che et al., 2018; Cao et al., 2018; Yoon et al., 2019) use gated mechanisms and hidden state propagation to capture intricate non-linear relationships for imputation, but iterative RNN is exceptionally time-consuming, and they usually struggle to leverage spatial interactions, making them prone to overfitting.

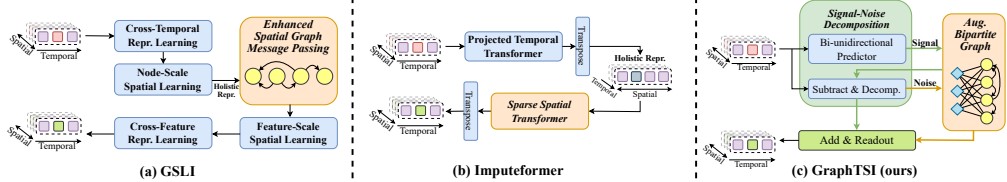

Figure 1: Structure comparison among GSLI, Imputeformer and GraphTSI

Recently, two promising imputation frameworks are emerging in the field: the graph-based methods (Yang et al., 2025b) and the transformer-based models (Nie et al., 2024). The graph-based models (overview in Figure 1(a)), with GSLI (Yang et al., 2025b) as representative, integrate spatial graph message passing layers to regularize information exchange. As for Transformer-based models (overview in Figure 1(b)), Imputeformer (Nie et al., 2024) stands out by incorporating a projected self-attention block and Fourier imputation loss to limit model expressivity against overfitting.

In spite of these breakthroughs, two critical limitations remain. The first is ***Insufficient Component Separation***. Multivariate time series can be represented as a composition of multiple components with distinct characteristics (Anderson, 2018; Cleveland & Cleveland, 1990). However, existing imputation methods treat each time series as a whole, directly imputing from observed inputs. While TIDER (LIU et al., 2023a) attempts to decompose time series into trend and seasonality components, it fails to capture interactions within noise, limiting its effectiveness. Moreover, time series decomposition is challenging in the presence of missing values. Conventional pre-processing decomposition commonly adopted in related fields (Zhou et al., 2022; Zeng et al., 2023; Wang et al., 2023) employs moving average or convolution, which exhibits large variance or even fails when long sequences of data are missing.

The second limitation is ***Linear Transformation of Spatial-Temporal Hidden States***. Spatial and temporal interactions differ in propagation and semantics (Yoon et al., 2019), making it essential to model them jointly yet distinctly for effective imputation. Existing methods follow the M-RNN (Yoon et al., 2019) structure, and stack interleaved spatial and temporal layers, via spatial graph message passing (Cini et al., 2021) and spatial attention (Nie et al., 2024), or employ spatial-temporal joint attention (Marisca et al., 2022; Wu et al., 2024). These methods forces the model to learn spatial and temporal representations that share the same feature space, leading to entangled or noisy features. Consequently, a key challenge lies in designing transformation layers that can effectively map between spatial and temporal representations, thereby capturing complex spatial-temporal interactions accurately.

Motivated by these limitations and the observation that a time series naturally exhibits two complementary components, with a signal component capturing predictable dynamics such as trend and seasonality, and a noise component reflecting unpredictable shocks, we propose GraphTSI, a **graph**-based **T**ime-**S**eries **I**mputation model with signal-noise decomposition (Figure 1c). The incomplete multivariate time series first passes through the *signal-noise decomposition block*, extracting signal and noise components. The two components further pass through the *augmented bipartite graph layer* to adaptively exchange spatial information, effectively capturing spatial-temporal interactions.

Specifically, for limitation 1, we propose a prediction-subtraction-based method for time series decomposition, grounded in the widely adopted data generation process (Anderson, 2018). It has two main steps: the prediction step and the subtraction step. With a new masked bi-unidirectional predictor, the prediction step actively predicts the signal component by utilizing either past or future information. This enforces signal prediction based on the learned temporal patterns rather than in-place filling, preventing information leakage from noise component. The subtraction step then compares the predicted signal component with the observed ground-truth to extract the noise component, thereby allowing separate and customized downstream processing of different components.

To address limitation 2, we construct an augmented bipartite graph in which structural spatial nodes and temporal nodes form two disjoint partitions, with spatial representations modeled as bipartite edges. To further regulate spatial interaction, we use augmented edges that directly connect pairs of spatial nodes within the spatial partition. Information is transmitted through three mechanisms: gather, propagate, and dispatch. The information is first gathered from the bipartite edges to derive spatial representations at each node. The introduced augmented edges facilitate direct information propagation among spatial nodes, effectively functioning as a spatial interaction layer. Finally, the

aggregated information is dispatched back onto the edges as updated temporal representations. This design enables flexible information exchange of spatial-temporal hidden states.

To sum up, our main contributions are as follows:

1. We propose a new method, GraphTSI, for multivariate time series imputation that integrates signal-noise decomposition with graph propagation.

2. We propose a prediction-subtraction-based method for decomposition via a masked bi-unidirectional prediction framework. It enables robust decomposition of signal components and noise components and remains effective even under extremely high missing rates.

3. We introduce an augmented bipartite graph for adaptive transformation between spatial and temporal representations, allowing more accurate component-specific feature interactions.

4. We conduct extensive experiments on 12 real-world multivariate time series datasets, achieving an average reduction in mean absolute error of 10.273% and 17.580% compared to GSLI and Imputeformer, respectively. Additional experiments on downstream tasks, ablation studies, extreme missing rates, robustness analysis, and case studies further solidify the effectiveness of our model.

## 2 PRELIMINARY

**Notations and concepts**. Consider a multivariate time series $\mathbf{X} \in \mathbb{R}^{N \times T \times C}$ comprising measurements from $N$ distinguishable sensors observed over $T$ time steps, where each sensor provides $C$ channels of data. Specifically, the $C$-dimensional vector $\mathbf{X}_{i,t} \in \mathbb{R}^C$ from sensor $i$ at time step $t$ constitutes an *observation*, where each scalar $\mathbf{X}_{i,t,c} \in \mathbb{R}$ denotes a *measurement* of channel $c$. Due to inevitable sensor failures in real-world deployments, completeness of observations is compromised. This is captured by a binary observation missing mask $\mathbf{M} \in \{0,1\}^{N \times T}$, where $\mathbf{M}_{i,t} = 0$ indicates a missing observation for sensor $i$ at time step $t$.

To enhance imputation robustness against missing observations, we follow previous works (Nie et al., 2024; Cini et al., 2021) and incorporate two categories of exogenous information: (1) Temporal covariates $\mathbf{E}^{\mathrm{T}} \in \mathbb{R}^{T \times d_{\mathrm{T}}}$ such as time-of-day or day-of-week indices, representing cyclical patterns in measurements; (2) Spatial exogenous graph $\mathcal{G}^{\mathrm{S}} = \langle \mathcal{V}^{\mathrm{S}}, \mathcal{E}^{\mathrm{S}} \rangle$ with $\left| \mathcal{V}^{\mathrm{S}} \right| = N$, derived from geographical distances or naive correlation matrices, representing stationary spatial relationships among sensors. (implementation detail for each dataset is in Section B.1)

However, directly imputing over the full temporal horizon (especially for $T \gg 10^5$) is computationally prohibitive. To address this, we follow previous works (Yi et al., 2016; Cao et al., 2018; Cini et al., 2021; Nie et al., 2024) and adopt a sliding window approach by sampling subseries of length $W$ ($W \ll T$) along the time axis. For a sliding window starting at time step $t$, we extract the following: (1) Ground truth $\mathbf{X}^{(t)} := \mathbf{X}_{:,t:t+W} \in \mathbb{R}^{N \times W \times C}$; (2) Missing mask $\mathbf{M}^{(t)} := \mathbf{M}_{:,t:t+W} \in \{0,1\}^{N \times W}$; (3) Partially observed input $\widetilde{\mathbf{X}}^{(t)} := \mathbf{M}^{(t)} \odot \mathbf{X}^{(t)}$, where $\odot$ denotes element-wise multiplication broadcasting over the channel dimension $c$; (4) Temporal covariates $\mathbf{E}^{\mathrm{T},(t)} := \mathbf{E}^{\mathrm{T}}_{t:t+W} \in \mathbb{R}^{W \times d_{\mathrm{T}}}$.

**Problem Formulation**. Given an incomplete multivariate time series $\widetilde{\mathbf{X}}^{(t)}$ and a missing mask $\mathbf{M}^{(t)}$, multivariate time series imputation aims to predict the unobserved entries in $\widetilde{\mathbf{X}}^{(t)}$ so that the imputed series $\widehat{\mathbf{X}}^{(t)}$ is as close as possible to the true series $\mathbf{X}^{(t)}$.

**Data Generating Process**. In the real world, the data generating process (DGP) of time series can vary widely across datasets and across time periods. We follow previous works (Blasques et al., 2020; Wu & Politis, 2024; Armillotta & Fokianos, 2023) and focus on two essential and complementary components commonly present in time series, signal and noise, to model the DGP. For each observation, we assume:

$$\mathbf{X}_{i,\tau} = f\left(\mathbf{X}_{:,<\tau}\right) + \varepsilon_{i,\tau} \tag{1}$$

where $f\left(\cdot\right)$ is a non-linear autoregressive function describing the predictable expected observation based on past information, which we refer to as the signal component. Conversely, $\mathrm{E}\left[\varepsilon_{i,\tau}\right] = 0$ corresponds to the exogenous noise component. Furthermore, related literature (Blasques et al., 2020; Wu & Politis, 2024; Armillotta & Fokianos, 2023) commonly adopts the following assumptions (see

Appendix A.3 for validity of these assumptions) regarding the statistical properties of $\varepsilon_{i,\tau}$:

$$\varepsilon_{i,\tau} \perp \mathbf{X}_{j,\omega}, \quad \forall j \in \{1,\ldots,N\}, \, \omega < \tau \tag{2}$$

$$\varepsilon_{i,\tau} \perp \varepsilon_{j,\omega}, \quad \forall j \in \{1,\ldots,N\}, \, \omega \neq \tau \tag{3}$$

$$\varepsilon_{i,\tau} \not\perp \varepsilon_{j,\tau}, \quad \forall j \in \{1,\ldots,N\} \tag{4}$$

The first assumption ensures each noise component to be independent of all prior observations. The second and third assumptions constrain the spatial-temporal interactions among noise components: those at different time steps are mutually independent, while contemporaneous ones may exhibit dependencies. In particular, noise at different time steps generally arises independently, whereas contemporaneous noise may arise from shared causes across variables, such as system-wide incidents or environmental disturbances.

## 3 RELATED WORK

Extensive methods are proposed for multivariate time series imputation, and can be broadly categorized into the following paradigms: 1) *Statistical Methods*. Early methods were mainly statistical, using measures such as weighted averages, medians, modes, or linear interpolation (Moritz & Bartz-Beielstein, 2017; Bashir & Wei, 2018). 2) *Matrix Factorization Methods*. Matrix factorization approaches leverage the inherent low-rank and repetitive patterns in time series to estimate missing values through linear subspace modeling. For example, FGTI (Yu et al., 2016) incorporates high-frequency filtering to better exploit residual information, while LATC (Chen et al., 2022) introduces temporal regularization for improved local consistency. 3) *RNN-based Methods*. Representative RNN methods include GRU-D (Che et al., 2018), BRITS (Cao et al., 2018), M-RNN (Yoon et al., 2019), and NADE (Berglund et al., 2015), which leverage the recurrent structure to successively integrate information across time steps with hidden states in an auto-regressive paradigm. M-RNN (Yoon et al., 2019) introduces a interleaving temporal-spatial approach for imputation. BRITS (Cao et al., 2018) extends RNN with bidirectional inference and masked regression to better aggregate information from both past and future observations. 4) *Generative Methods*. Generative methods (Yang et al., 2024; Islam et al., 2025) are also widely used to capture the data distribution. 5) *Transformer-based Methods*. The transformer architecture (Vaswani et al., 2017) has demonstrated substantial performance gains in time series imputation by enabling global receptive fields via the self-attention mechanism. SAITS (Du et al., 2023) introduces a diagonally masked self-attention scheme. SPIN (Marisca et al., 2022) utilizes multiple sparse attention patterns to model intra-sensor and inter-sensor dependencies, which forces all states to share the same representation space, hindering preformance. Imputeformer (Nie et al., 2024) exploits the low-rank structure of time series by employing projected temporal attention and spatial embedding attention. 6) *Graph-based Methods*. Graph-based approaches model multivariate time series imputation as a node or edge feature completion task on spatial graphs, where sensors are represented as nodes, and relationships as edges. For instance, STCAGCN (Nie et al., 2023) focuses on traffic data and dynamically constructs graphs based on additional speed sensor information. FC-GNN (Satorras et al., 2022) leverages spatial graph to capture intra-sensor dependencies. GACN (Ye et al., 2021) employs graph convolution layers to refine spatial encoding. STAR (Liang et al., 2023) merges temporal features into spatial features for node propagation. GRIN (Cini et al., 2021) integrates GRU to iteratively aggregate temporal information with spatial message passing for spatial-temporal fusion. GSLI (Yang et al., 2025b) introduces multi-scale graph structure learning to enhance spatial correlations. Graph-based methods are also widely used in tabular data (Wang et al., 2025a).

## 4 METHODS

The overview of GraphTSI is shown in Figure 2. Specifically, given an incomplete multivariate time series, GraphTSI first maps observations into information-rich vectors via the missing-aware input embedding (Section 4.1). Inside the signal-noise decomposition block (Section 4.2), a bi-unidirectional predictor estimates univariate signals from these embeddings, followed by an augmented bipartite graph (Section 4.3) for spatial interactions, yielding the updated signal components. Then, signal and noise components are separated by subtracting the predicted signal from observed ground truth, followed by an augmented bipartite graph for missing noise imputation. Finally, imputation is generated via MLP readout of aggregated signal and noise components.

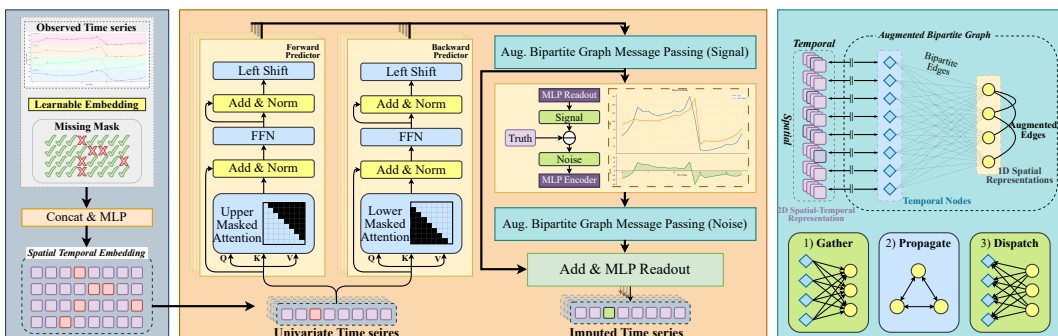

Figure 2: The architecture overview of the model.

## 4.1 MISSING AWARE INPUT EMBEDDING

Conventional embedding methods typically employ an MLP for feature extraction and concatenate temporal positional embedding afterwards. However, this approach fails to differentiate between missing and observed measurements. Therefore, we adapt from (Lipton et al., 2016) and use missing-aware input embedding through concatenating temporal positions, missing masks and observed ground truth prior the MLP. Formally, given $\widetilde{\mathbf{X}}^{(t)}$, $\mathbf{M}^{(t)}$ and $\mathbf{E}^{\mathrm{T},(t)}$:

$$\mathbf{h}_{i,\tau}^{(0)} = \mathrm{MLP}\left(\widetilde{\mathbf{X}}_{i,\tau}^{(t)} \middle\| \mathbf{M}_{i,\tau}^{(t)} \middle\| \mathbf{E}_{i,\tau}^{\mathrm{T},(t)}\right) \tag{5}$$

where $i \in [1, N]$ indexes the sensors and $\tau \in [1, W]$ indexes the time steps. As for the exogenous temporal covariates, we followed previous works (Vaswani et al., 2017; Nie et al., 2024) with a sinusoidal time-of-day encoding for each observation.

## 4.2 SIGNAL-NOISE DECOMPOSITION

To decompose signal and noise in time series data, we propose a prediction-subtraction method that first predicts the predictable signal component from past observations and then subtracts the signal component from the observed ground truth to extract the noise component. Theorem 2 shows that training on raw observations is equivalent to training on their predictable signal, ensuring that the predictor captures the signal without being affected by noise, and thus validating our prediction–subtraction-based approach.

**Bi-unidirectional Prediction Step**. We adopt a bi-unidirectional design with two independent unidirectional predictors in opposite directions. This approach preserves identifiability between signal and noise while enhancing boundary prediction through aggregating complementary evidence from both directions. For clarity, we will describe only the forward unidirectional predictor; the reverse direction is implemented analogously.

The forward predictor estimates the expected univariate signal component for each sensor over the time axis. We adopt a standard Transformer (Vaswani et al., 2017) as the backbone due to its parallelization efficiency and simplicity. Specifically, we apply a triangularly masked self-attention mechanism across the temporal dimension:

$$\mathbf{h}_i^{(l),\mathrm{attn,f}} = \mathrm{MaskedSelfAttn}\left(\mathbf{h}_i^{(l-1)}, \mathbf{h}_i^{(l-1)}, \mathbf{h}_i^{(l-1)}\right)$$
$$= \mathrm{MaskedSoftmax}\left(\frac{\mathbf{h}_i^{(l-1)}\mathbf{W}_Q\mathbf{W}_K^\top\mathbf{h}_i^{(l-1),\top}}{\sqrt{d_\mathrm{h}}}\right)\mathbf{h}_i^{(l-1)}\mathbf{W}_V \tag{6}$$

where $\mathbf{W}_Q, \mathbf{W}_K, \mathbf{W}_V \in \mathbb{R}^{d_h \times d_h}$ are learnable projection matrices, and $\mathrm{MaskSoftmax}\,(\cdot)$ applies an upper-triangular masking so that each time step only attend to previous and current observations. The resulting hidden states then pass through residually connected layer normalization and feed

forward layers to increase expressivity:

$$\mathbf{h}_i^{(l),\text{norm,f}} = \text{LayerNorm}\left(\mathbf{h}_i^{(l-1)} + \mathbf{h}_i^{(l),\text{attn,f}}\right) \tag{7}$$

$$\mathbf{h}_i^{(l),\text{ffn,f}} = \text{LayerNorm}\left(\mathbf{h}_i^{(l),\text{norm,f}} + \text{FFN}\left(\mathbf{h}_i^{(l),\text{norm,f}}\right)\right) \tag{8}$$

However, residual connections and triangular self-attention allow present-step information to leak into each hidden state, potentially causing trivial auto-encoding solutions. To ensure that hidden states only incorporates historical information, we shift the features one step forward in time:

$$\widetilde{\mathbf{h}}_{i,:}^{(l),\text{signal,f}} = \mathbf{W}_{\text{fill,f}} \middle\| \mathbf{h}_{i,1:W}^{(l),\text{ffn,f}} \tag{9}$$

where $\mathbf{W}_{\text{fill,f}} \in \mathbb{R}^{N \times d_h}$ is a learnable hidden state for the first time step.

**Augmented Bipartite Graph for Signal**. This signal hidden state only contains univariate information within each time series, so we pass them through the augmented bipartite graph transformation for spatial interactions and retrieve the full multivariate signal features: $\mathbf{h}_{i,:}^{(l),\text{signal,f}} = \text{AugBipGraph}_{\text{signal}}\left(\widetilde{\mathbf{h}}_{i,:}^{(l),\text{signal,f}}\right)$, where $\text{AugBipGraph}_{\text{signal}}(\cdot)$ represents the customized signal augmented bipartite graph in Section 4.3.2.

**Subtraction Step**. To separate the two components, we first extract our predicted signal component from information-rich hidden state through an MLP readout: $\widehat{\mathbf{X}}_{i,\tau}^{(l),\text{signal,f}} = \text{MLP}\left(\mathbf{h}_{i,\tau}^{(l),\text{signal,f}}\right)$. Subsequently, the exogenous noise component is extracted from observed measurements by subtracting the predicted signal from observed ground truth:

$$\widetilde{\mathbf{X}}_{i,\tau}^{(l),\text{noise,f}} = \mathbf{M}_{i,\tau}^{(t)} \times \left(\widetilde{\mathbf{X}}_{i,\tau}^{(t)} - \widehat{\mathbf{X}}_{i,\tau}^{(l),\text{signal,f}}\right) \tag{10}$$

Missing entries where $\mathbf{M}_{i,\tau}^{(t)} = 0$ are set to 0. Now, extracted noise component in both directions are concatenated and passes through another missing-aware embedding to produce complete noise features for downstream imputation: $\widetilde{\mathbf{h}}_{i,\tau}^{(l),\text{noise}} = \text{MLP}\left(\widetilde{\mathbf{X}}_{i,\tau}^{(l),\text{noise,f}} \middle\| \widetilde{\mathbf{X}}_{i,\tau}^{(l),\text{noise,b}} \middle\| \mathbf{M}_{i,\tau}^{(t)}\right)$. Here, $\widetilde{\mathbf{h}}_{i,\tau}^{(l),\text{noise}}$ is the initial noise embedding and acts as the input for imputing the missing noise components through the augmented bipartite graph.

## 4.3 Augmented Bipartite Graph Transformation

We construct an augmented bipartite graph to capture both spatial-temporal representation transformation and spatial dependencies among sensors. The graph consists of two types of nodes: temporal nodes and spatial nodes, with bipartite edges in between to form a bipartite structure and additional augmented edges for spatial information propagation. Formally, the bipartite graph is defined as: $\mathcal{G}^{\text{bip}} = \left\langle \mathcal{V}^{\text{T}} \bigcup \mathcal{V}^{\text{S}}, \mathcal{E}^{\text{bip}} \right\rangle$, where $\mathcal{V}^{\text{T}} = \{v_1^t, \ldots, v_W^t\}$ denotes the set of temporal nodes; $\mathcal{V}^{\text{S}} = \{v_1^s, \ldots, v_N^s\}$ the set of spatial nodes; and $\mathcal{E}^{\text{bip}} = \{(v_i^t, v_\tau^s)|\mathbf{M}_{i,\tau}^{(t)} = 1\}$ the set of bipartite edges, corresponding to each spatial-temporal observations. In addition, we introduce the spatial exogenous graph $\mathcal{G}^{\text{S}} = \left\langle \mathcal{V}^{\text{S}}, \mathcal{E}^{\text{S}} \right\rangle$ connecting pairs of spatial nodes as the augmented propagation graph. The initial representations $\widetilde{\mathbf{h}}_{i,\tau}^{(l),\text{signal}}$ or $\widetilde{\mathbf{h}}_{i,\tau}^{(l),\text{noise}}$ computed for signal components and noise components respectively are naturally mapped to the edge embedding of corresponding bipartite edges $(v_i^t, v_\tau^s)$, such that each edge embeds all relevant information associated with observation $\mathbf{X}_{i,\tau}^{(t)}$. This construction allows the model to flexibly convert temporal and spatial representation via edge-node message propagation within $\mathcal{G}^{\text{bip}}$, and present spatial interactions through node-node message propagation within $\mathcal{G}^{\text{S}}$. We first introduce a generalized message passing procedure on the augmented graph, which can be customized for signal and noise to better fit their characteristics.

### 4.3.1 Generalized Message Passing Procedure

The proposed message passing framework consists of three mechanisms: 1) *Gather*; 2) *Propagate*; 3) *Dispatch*. These three mechanisms transform temporal presentation to spatial representation, exchange spatial information and transform them back to temporal representations. Specifically:

**Gather**. For each spatial node $i$, the gather function aggregates temporal representations from bipartite edges associated with sensor $i$ to form a spatial representation:

$$\mathbf{h}_i^{(l),\text{gather}} = \text{Gather}\left(\left\{\widetilde{\mathbf{z}}_{i,\tau}^{(l)}\Big|(v_i^{\text{t}}, v_\tau^{\text{s}}) \in \mathcal{E}^{\text{bip}}\right\}\right) \tag{11}$$

where $\widetilde{\mathbf{z}}_{i,\tau}^{(l)}$ is the input bipartite edge embedding (i.e. $\widetilde{\mathbf{h}}_{i,\tau}^{(l),\text{signal}}$ for signal and $\widetilde{\mathbf{h}}_{i,\tau}^{(l),\text{noise}}$ for noise).

**Propagate**. Spatial nodes propagate information the spatial exogenous graph $\mathcal{G}^{\text{S}}$:

$$\mathbf{h}_i^{(l),\text{prop}} = \text{FFN}\left(\sum_{j \in \text{N}(i;\mathcal{E}^{\text{S}})} \mathbf{W}_{i,j}\mathbf{h}_i^{(l),\text{gather}}\right) \tag{12}$$

where $\text{N}\left(i;\mathcal{E}^{\text{S}}\right)$ denotes all neighbors of node $v_i^{\text{s}}$ connected via the given spatial exogenous graph edge set $\mathcal{E}^{\text{S}}$; $\mathbf{W}_{i,j}$ specifies the learnable edge weight for that spatial graph, allowing signal and noise to share the same graph structure, but adopt independent relationships through learnable parameters. To provide this flexibility, we utilize a learnable characteristic matrix $\mathbf{E}^{\text{S}} \in \mathbb{R}^{N \times d_{\text{S}}}$, and compute the edge weights as: $\mathbf{W} = \text{Softmax}\left(\mathbf{E}^{\text{S}}\mathbf{W}^{(l)}\mathbf{E}^{\text{S},\top}, \dim = 1\right)$, where $\mathbf{W}^{(l)} \in \mathbb{R}^{d_{\text{S}} \times d_{\text{S}}}$ is a component-specific projection matrix for signal and noise.

**Dispatch**. The updated spatial node representations are distributed back to bipartite graph edges as temporal representations for inference: $\mathbf{z}_{i,\tau}^{(l)} = \text{Dispatch}\left(\mathbf{h}_i^{(l),\text{prop}}\right)$ where $\mathbf{z}_{i,\tau}^{(l)}$ is the updated temporal representation ($\mathbf{h}_{i,\tau}^{(l),\text{signal}}$ for signal and $\mathbf{h}_{i,\tau}^{(l),\text{noise}}$ for noise).

**Readout**. Finally, we form a combined representation by concatenating both updated signal features with the updated noise features, from which an MLP readout yields the final imputation: $\widehat{\mathbf{X}}_{i,\tau}^{(t)} = \text{MLP}\left(\mathbf{h}_{i,\tau}^{(l),\text{signal,f}}\Big\|\mathbf{h}_{i,\tau}^{(l),\text{signal,b}}\Big\|\mathbf{h}_{i,\tau}^{(l),\text{noise}}\right)$

The training objective of GraphTSI follows previous works (Du et al., 2023), combining both the observed reconstruction loss and the masked imputation loss (see Section B.3 for more details).

### 4.3.2 Customization for Signal and Noise Components

**Customization for Signal Components**. Signal components tend to exhibit temporal redundancy with smooth transitions (Nie et al., 2024). To capture these properties, we employ a projected attention for the gather and dispatch functions:

$$\text{Gather}_{\text{signal}}\left(\left\{\widetilde{\mathbf{h}}_{i,\tau}^{(l),\text{signal}}\Big|(v_i^{\text{t}}, v_\tau^{\text{s}}) \in \mathcal{E}^{\text{bip}}\right\}\right) = \text{Flatten}\left(\text{Attn}\left(\mathbf{W}_{\text{proj}}, \mathbf{h}_{i,:}^{(l),\text{shift}}, \widetilde{\mathbf{h}}_{i,\tau}^{(l),\text{signal}}\right)\right) \tag{13}$$

where $\text{Attn}(\cdot)$ denotes the standard attention mechanism from (Vaswani et al., 2017); while $\mathbf{W}_{\text{proj}} \in \mathbb{R}^{H \times d_{\text{h}}}$ is the learnable query matrix that projects the original $W$ time steps to $H$ ($H \ll W$) heads in the latent space; and $\text{Flatten}(\cdot) : \mathbb{R}^{H \times d_{\text{h}}} \mapsto \mathbb{R}^{Hd_{\text{h}}}$ stacks all heads into a single high dimensional vector as the spatial representation. Similarly, to recover signal features during dispatch, we perform the reverse process: un-flatten the updated spatial representations, then apply an attention to project them back to temporal representations for each time step:

$$\text{Dispatch}_{\text{signal}}\left(\mathbf{h}_i^{(l),\text{signal,prop}}\right) = \text{Attn}\left(\widetilde{\mathbf{h}}_{i,\tau}^{(l),\text{signal}}, \mathbf{W}_{\text{proj}}, \text{Unflatten}\left(\mathbf{h}_i^{(l),\text{signal,prop}}\right)\right) \tag{14}$$

where $\text{Unflatten}(\cdot) : \mathbb{R}^{Hd_{\text{h}}} \mapsto \mathbb{R}^{H \times d_{\text{h}}}$ is the inverse of flatten. This encoder-decoder structure enables compact latent encoding and robust signal recovery, thereby reducing noise and increasing computational efficiency specifically for signal components.

**Customization for Noise Components**. Noise components capture sudden, non-repetitive features, with little temporal redundancy among the inputs. Therefore, we preserve all information by performing direct flatten and un-flatten operations without additional processing. Formally:

$$\text{Gather}_{\text{noise}}\left(\left\{\widetilde{\mathbf{h}}_{i,\tau}^{(l),\text{noise}}\Big|(v_i^{\text{t}}, v_\tau^{\text{s}}) \in \mathcal{E}^{\text{bip}}\right\}\right) = \text{Flatten}\left(\widetilde{\mathbf{h}}_{i,\tau}^{(l),\text{noise}}\right) \tag{15}$$

$$\text{Dispatch}_{\text{noise}}\left(\mathbf{h}_i^{(l),\text{noise,prop}}\right) = \text{Unflatten}\left(\mathbf{h}_i^{(l),\text{noise,prop}}\right) \tag{16}$$

where $\text{Flatten}(\cdot) : \mathbb{R}^{W \times d_{\text{h}}} \mapsto \mathbb{R}^{Wd_{\text{h}}}$ and $\text{Unflatten}(\cdot) : \mathbb{R}^{Wd_{\text{h}}} \mapsto \mathbb{R}^{W \times d_{\text{h}}}$ act as dimension-preserving transformations for maximal retention of abrupt exogenous noise event information.

### 4.4 ANALYSIS

**Superiority of GraphTSI**. In Theorem 1, we show that conventional spatial-temporal interconversion and interaction layers arise as special cases of our framework. The proof is in Section A.2.

**Theorem 1.** *The spatial-temporal information exchange mechanism employed by existing state-of-the-art multivariate imputation methods (i.e., GRIN, GSLI, and Imputeformer) can be regarded as a special case of our proposed method.*

**Time Complexities Analysis**. The missing aware input embedding has a time complexity of $O(WNCd_{\mathrm{h}})$. The bi-unidirectional predictors adopt standard transformer architecture, leading to a time complexity of $O(W^2Nd_{\mathrm{h}})$. The augmented bipartite graph gathers and dispatches information with a time complexity of $O(WHNd_{\mathrm{h}})$ for signal and $O(WNd_{\mathrm{h}})$ for noise component. Propagation is $O(|\mathcal{E}^{\mathrm{S}}|Hd_{\mathrm{h}})$ for signal and $O(|\mathcal{E}^{\mathrm{S}}|Nd_{\mathrm{h}})$ for noise, where $H$ is the number of latent heads, and $|\mathcal{E}^{\mathrm{S}}|$ is the number of edges in exgenous spatial graph. The overall time complexity is $O\left(WNd_{\mathrm{h}}(W + H + C) + |\mathcal{E}^{\mathrm{S}}|d_{\mathrm{h}}(H + N)\right)$, which scales quadratically with window sizes. Additional results on model efficiency can be found in Figure 6.

## 5 EXPERIMENTS

### 5.1 EXPERIMENTAL SETUP

**Datasets**. Following previous works (Nie et al., 2024; Cini et al., 2021), we use 9 real-world public datasets, including AQI36, AQI, METR-LA, PEMS-BAY, PEMS03, PEMS04, PEMS07, PEMS08, and HAR (Details in Section B.1). We adopt MAE and RMSE as evaluation metrics, with their definitions and the dataset masking strategy detailed in Section B.2.

**Baselines**. We compare GraphTSI with nine baselines spanning various categories: (1) **Average**: Statistical average of each sensor; (2) **GRIN** (Cini et al., 2021): A graph-based model combining GRU with message passing for imputation; (3) **GSLI** (Yang et al., 2025b): A graph-based model with multi-scale graph structure learning for imputation; (4) **BRITS** (Cao et al., 2018): A bidirectional RNN model for imputation; (5) **TIDER** (LIU et al., 2023a): A matrix factorization method with explicit modeling for different time series components; (6) **adaTIDER** (Liu et al.): A matrix factorization method that incorporates adaptive cross-channel dependency modeling and multi-period seasonality representations; (7); **LCR** (Chen et al.): An efficient low-rank Laplacian convolutional representation model. (8) **UnIMP** (Wang et al., 2025a): A SOTA tabular imputation method; (9) **SAITS** (Du et al., 2023): A transformer-based model with diagonally masked attention blocks; (10) **Imputeformer** (Nie et al., 2024): A low-rank induced transformer-based model.

**Implementation Details**. We follow Imputeformer (Nie et al., 2024) for experimental settings, with a default point missing rate of $25\%$, block missing rate of $0.5\%$, and window size of $24$. The model is trained for a maximum of 300 epochs with patience of 30. All tests are done with seed 2. Experiments are conducted on a server equipped with a 24 GB NVIDIA RTX 4090 GPU.

### 5.2 RESULTS

**Overall Results**. We evaluate imputation error across all datasets, with MAE results shown in Table 1, and RMSE results shown in Table 3. As we can see, the transformer-based SOTA Imputeformer performs well in traffic and HAR datasets, where time series exhibit strong seasonal repetition, but performs poorly when such seasonality is absent. Graph-based SOTA GSLI performs well on PEMS datasets with multiple channels per sensor. Our GraphTSI, with signal-noise decomposition design and augmented bipartite graph, performs best in all scenarios, achieving an average MAE improvement of $17.215\%$ over Imputeformer and $9.557\%$ over GSLI, respectively.

**Ablation Study**. This experiment conducts an ablation study on key components of GraphTSI. Specifically, we consider three ablation setups: 1) *w/o Noise*: We remove the Signal-Noise decomposition layer and directly decode results from signal features $\mathbf{h}_{i,:}^{(l),\mathrm{signal,f}}$ and $\mathbf{h}_{i,:}^{(l),\mathrm{signal,b}}$; 2) *Bidirectional*: We employ a bidirectional approach instead of the bi-unidirectional approach for the prediction step; 3) *w/o Graph*: We disable spatial interactions for augmented bipartite graph message passing by setting the adjacency matrix to a unit matrix. Ablations are performed on AQI, PEMS08, and HAR to maximize domain diversity. Results summarized in Figure 3 indicate that the three techniques bring an average performance gain of $15.648\%$, $5.561\%$, and $43.788\%$, respectively.

| Dataset | Mask | Avg. | GRIN | GSLI | BRITS | TIDER | adaTIDER | LCR | UnIMP | SAITS | Imputeformer | GraphTSI | |
|---|---|---|---|---|---|---|---|---|---|---|---|---|---|
| AQI36 | Simulated | 52.6813 | 12.5883 | 12.8262 | 14.1221 | 24.2235 | 47.7634 | 54.7605 | 19.4776 | 16.3931 | 13.9852 | 10.3630 | -17.678% |
| AQI | Simulated | 39.8759 | 14.7626 | 15.8241 | 19.8495 | 32.7432 | 31.9666 | 33.3918 | 21.1230 | 21.6905 | 16.3739 | 13.2298 | -10.383% |
| METR-LA | Point | 15.0783 | 1.8863 | 1.7012 | 3.7869 | 9.2683 | 10.1894 | 9.5630 | 2.2684 | 3.0864 | 1.6891 | 1.6339 | -3.268% |
| | Block | 15.1521 | 2.5720 | 2.3399 | 3.9786 | 9.6889 | 10.9506 | 9.3800 | 3.3800 | 3.7550 | 2.3017 | 2.2353 | -2.885% |
| PEMS-BAY | Point | 5.3772 | 0.6570 | 0.6167 | 1.9095 | 3.9580 | 4.0253 | 4.2214 | 0.8945 | 1.6308 | 0.5870 | 0.5692 | -1.980% |
| | Block | 5.4192 | 1.0668 | 1.0749 | 1.9474 | 3.9384 | 3.9962 | 4.1967 | 1.4736 | 1.9188 | 1.0253 | 0.9473 | -7.608% |
| PEMS03 | Point | 85.2738 | 9.2689 | 7.6272 | 12.6884 | 34.4900 | 37.5310 | 36.4490 | 11.1235 | 15.2878 | 7.4442 | 7.0871 | -4.797% |
| | Block | 85.8496 | 11.6494 | 9.1688 | 12.2455 | 31.8528 | 41.1549 | 33.0797 | 15.4294 | 15.4563 | 8.8392 | 8.1013 | -8.348% |
| PEMS04 | Point | 36.4437 | 5.4738 | 2.6111 | 6.7712 | 15.2647 | 16.3088 | 15.2068 | 6.0129 | 8.1783 | 5.6620 | 2.5512 | -2.294% |
| | Block | 36.6491 | 7.9954 | 5.0697 | 7.4038 | 15.0324 | 15.6130 | 15.2393 | 7.0645 | 8.4292 | 6.3202 | 4.5507 | -10.237% |
| PEMS07 | Point | 121.8156 | 11.7727 | 10.1087 | 23.9394 | 53.4188 | 54.5748 | 46.5116 | 16.7349 | 27.4032 | 10.6516 | 10.0745 | -0.338% |
| | Block | 122.3093 | 14.7600 | 13.1901 | 23.7428 | 51.2491 | 54.3342 | 46.7072 | 20.0892 | 28.2561 | 13.8238 | 12.7600 | -3.261% |
| PEMS08 | Point | 31.1824 | 4.2496 | 1.9014 | 5.5024 | 13.5194 | 15.0606 | 12.6943 | 4.5164 | 6.8455 | 4.0626 | 1.8007 | -5.296% |
| | Block | 31.3436 | 7.0010 | 4.1167 | 5.9880 | 13.2764 | 14.7039 | 12.5099 | 6.5420 | 7.3250 | 4.6939 | 3.6018 | -12.508% |
| HAR | Point | 0.2327 | D.N.F. | 0.0276 | 0.0481 | 0.1584 | 0.1288 | 0.1712 | 0.0640 | 0.0545 | 0.0283 | 0.0249 | -9.619% |
| | Block | 0.2284 | D.N.F. | 0.0325 | 0.0380 | 0.1219 | 0.0722 | 0.1259 | 0.1660 | 0.0516 | 0.0277 | 0.0237 | -14.440% |
| ETTm1 | Point | 5.9672 | 0.7907 | 0.4685 | 0.5749 | 2.6206 | 2.1944 | 3.2588 | D.N.F. | 0.4805 | 0.4779 | 0.4469 | -4.610% |
| | Block | 3.6908 | D.N.F. | 0.4991 | 0.5822 | 2.5922 | 2.1853 | 3.2951 | D.N.F. | 0.4483 | 0.5763 | 0.4179 | -6.781% |
| ETTm2 | Point | 6.4075 | D.N.F. | 0.3539 | 0.8668 | 4.1097 | 3.2316 | 5.3046 | D.N.F. | 0.3869 | 0.3956 | 0.2752 | -22.238% |
| | Block | 6.3221 | D.N.F. | 0.8787 | 1.4765 | 4.1759 | 3.1788 | 5.2098 | D.N.F. | 0.8460 | 1.2310 | 0.7001 | -17.246% |
| Elergone | Point | 244.1358 | D.N.F. | 27.3681 | 85.3951 | 155.5574 | 158.5652 | 165.6583 | 34.9038 | 72.5428 | 27.7954 | 27.2700 | -0.358% |
| | Block | 241.0613 | D.N.F. | 44.7311 | 82.2766 | 147.5384 | 150.2892 | 154.7714 | 67.3998 | 68.9966 | 41.3560 | 40.5752 | -1.888% |
| *Average Improvement* | | -88.050% | -27.147% | -10.273% | -47.648% | -78.350% | -79.477% | -80.247% | -44.039% | -45.890% | -17.580% | — | |

Table 1: MAE results. The final column shows average gain over second best method, and the final row shows average gain over each model. D.N.F. indicates not finished within 8 hours for ETTm1, ETTm2 and Elergone datasets and not finished within 48 hours for other datasets.

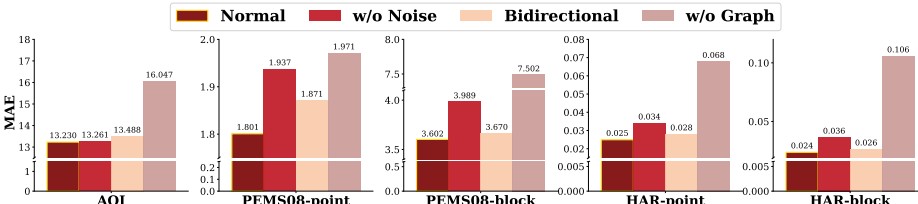

Figure 3: Ablation study on different datasets

**Downstream Task**. In this experiment, we evaluate the downstream classification performance on the HAR dataset. We employ a simple MLP layer as our classifier for all models, including results from ground-truth datasets (with no missing observations). Figure 4(a) presents the classification results. This shows that improved imputation quality can improve downstream task accuracy, with GraphTSI being outperforming other models both in point missing and block missing scenarios.

**Different Missing Rates**. We compare the performance of GraphTSI against SOTA models, including Imputeformer and GSLI, under various missing rates in Figure 4(b). More results are in Section C.2. Generally, performance degrades for all models under extremely sparse data. GraphTSI degrades slower compared to other models. Specifically, GraphTSI achieves 4.597% and 83.197% improvement over GSLI and Imputeformer, respectively, when the missing rate is 95%.

**Hyperparameter Analysis**. We investigate two hyperparameters: window size $W$ and model layers $L$. Results are shown in Figure 4(c) and Figure 4(d), respectively. More results are in Section C.3. As shown, GraphTSI produces stable imputation results across window sizes and model layers.

**Case Studies**. We demonstrate model explainability using intermediate results on the AQI36 dataset. Figure 5(a) presents visualization for the decomposed signal, noise, and model imputation. In the imputed region, sensors 4 and 13 exhibit similar downward signal components, while noise components show high variability. This separation demonstrates GraphTSI's capability to model the

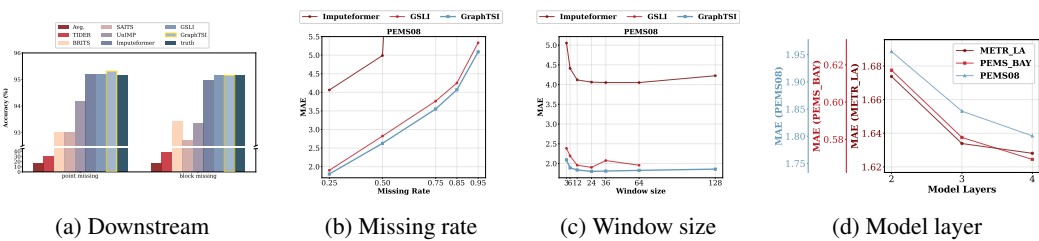

| (a) Downstream | (b) Missing rate | (c) Window size | (d) Model layer |
|---|---|---|---|

Figure 4: Results of downstream classification, different missing rates and hyper-parameters.

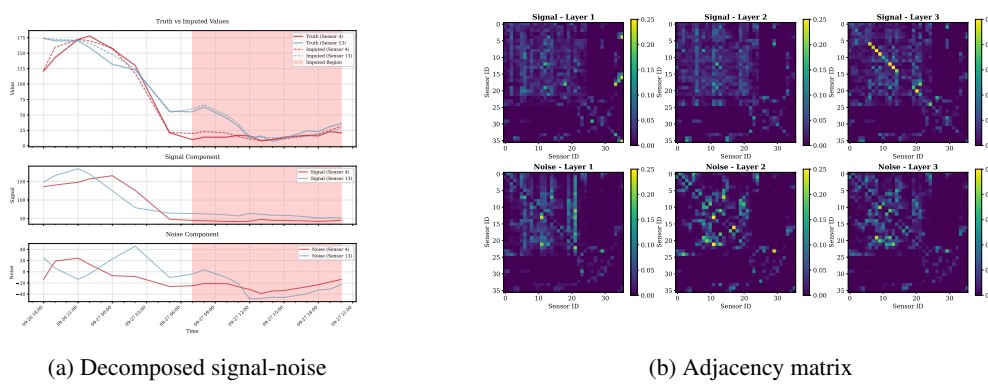

(a) Decomposed signal-noise

(b) Adjacency matrix

Figure 5: Case Study of Decomposed Signal-Noise and Adjacency Matrix.

distinct spatial interaction of signal and noise components with component-specific adjacency matrices. Further evidence (Figure 5(b)) visualizes learned adjacency weights of the augmented bipartite graph, where we can see distinct relationships in the upper-left corner: Signal components show two densely connected sub-regions, whereas noise interactions reveal complex sensor relationships.

**Model Efficiency**. In this experiment, we report the average inference speed and imputation error of different models to compare the efficiency of our model against other methods. Results are shown in Figure 6. Relative MAE is defined as the ratio between a model's MAE and the lowest MAE achieved by any model on the same dataset. On all the datasets, GraphTSI steadily delivers best imputation performance while maintaining inference speeds comparable to other strong baselines.

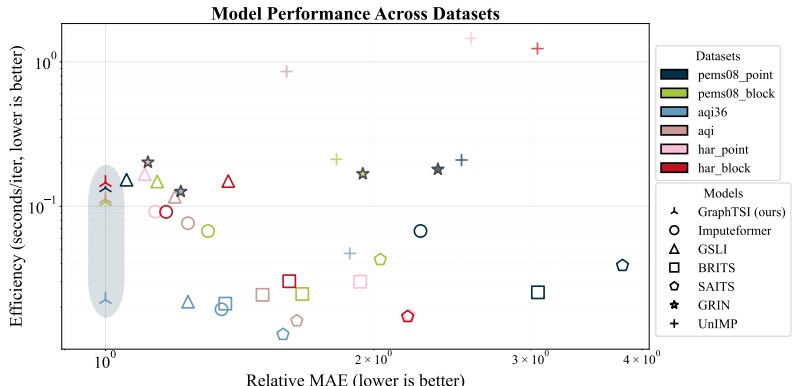

Figure 6: Average inference time and relative imputation error of different models.

# 6 CONCLUSION

In this paper, we propose GraphTSI, a graph-based multivariate time series imputation method. By leveraging a prediction–subtraction framework that decomposes each series into predictable signals and exogenous noise, and propagating information through an augmented bipartite graph for adaptive spatial–temporal representation, GraphTSI effectively captures essential information for accurate imputation. Experiments on 9 real-world datasets show the consistent superiority of GraphTSI.

## 7 REPRODUCIBILITY STATEMENT

We have taken multiple steps to facilitate reproducibility. Implementation details, including default missing rates, window sizes, training epochs, and training seed are documented in Section 5.1. We release anonymized code, scripts and configuration files for training, evaluating, as well as reproducing all reported tables at the following anonymous repository: https://anonymous.4open.science/r/timeseries-imputation-2E5B/README.md. Specific dataset usage and pre-processing steps are disclosed in Appendix B.1 and Appendix B.2.

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

# A    STATEMENTS AND PROOFS

## A.1    PROOF FOR EQUIVALENCE IN TRAINING OBJECTIVES

With our modeling of the DGP in Section 2, we can prove that the following theorem holds:

**Theorem 2** (Equivalence of model training target). *With a DGP in the from of Section 2, model trained under objective:*

$$\min_{\mathcal{W}} \sum_{i \in [1,N], \tau \in [1,W]} \left[ \text{NN} \left( \{ \mathbf{X}_{:,<\tau} \} ; \mathcal{W} \right)_i - \mathbf{X}_{i,\tau} \right]^2$$

*is equivalent to training under objective:*

$$\min_{\mathcal{W}} \sum_{i \in [1,N], \tau \in [1,W]} \left[ \text{NN} \left( \{ \mathbf{X}_{:,<\tau} \} ; \mathcal{W} \right)_i - f(\mathbf{X}_{:,<\tau})_i \right]^2$$

Therefore, we can employ a prediction-subtraction-based approach to decompose signal and noise components of each timeseries. Note that the signal component here stands for the expected value of each measurement based on past information, or namely $f(\mathbf{X}_{:,<\tau})$, and noise represents the remaining exogenous shock disturbance.

*Proof.* Consider a model training process where we update parameters $\mathcal{W}$ through optimizing the objective function:

$$\min_{\mathcal{W}} \sum_{i \in [1,N], \tau \in [1,W]} \left[ \text{NN} \left( \{ \mathbf{X}_{:,<\tau} \} ; \mathcal{W} \right)_i - \mathbf{X}_{i,\tau} \right]^2$$

Then, under the data generating process in Section 2, we substitute the iterative generating function into our objective:

$$\min_{\mathcal{W}} \sum_{i \in [1,N], \tau \in [1,W]} \left[ \text{NN} \left( \{ \mathbf{X}_{:<\tau} \} ; \mathcal{W} \right)_i - f(\mathbf{X}_{:,<\tau})_i - \varepsilon_{i,\tau} \right]^2$$

Let $\nu_{i,\tau}(\mathcal{W}) := \text{NN} \left( \{ \mathbf{X}_{:,\tau} \} ; \mathcal{W} \right)_i - f(\mathbf{X}_{:,<\tau})_i$, we can expand and simplify the square terms as:

$$\min_{\mathcal{W}} \sum_{i \in [1,N], \tau \in [1,W]} \nu_{i,\tau}^2 - 2\nu_{i,\tau}\varepsilon_{i,\tau} + \varepsilon_{i,\tau}^2$$

Now, we analyze each component: 1) The target objective term $\sum \nu_{i,\tau}^2$; 2) the cross term $\sum -2\nu_{i,\tau}\varepsilon_{i,\tau}$; 3) A constant squared noise term $\sum \varepsilon_{i,\tau}^2$. The third term can be omitted for it is a constant against our parameters. As for the cross term, we compute:

$$\mathbb{E} \left[ \frac{\partial}{\partial w} (-2\nu_{i,\tau}\varepsilon_{i,\tau}) \right] = -2\mathbb{E} \left[ \varepsilon_{i,\tau} \frac{\partial \nu_{i,\tau}}{\partial w} \right] = -2\mathbb{E}[\varepsilon_{i,\tau}] \mathbb{E} \left[ \frac{\partial \nu_{i,\tau}}{\partial w} \right] = 0$$

which implies the expected gradient for the second term vanishes because noise is independent of all previous observations. Therefore, on the whole, the only term contributing a non-zero expected gradient to our objective function would be the first term. Effectively, we are optimizing on

$$\min_{\mathcal{W}} \sum_{i \in [1,N], \tau \in [1,W]} \left[ \text{NN} \left( \{ \mathbf{X}_{:,<\tau} \} ; \mathcal{W} \right)_i - f(\mathbf{X}_{:,<\tau})_i \right]^2$$

Therefore, under the data generating process, the prediction-based decomposition can learn to fit on expected future signal though the optimization function contains noise. □

## A.2    LIMITATION OF PRIOR SPATIAL-TEMPORAL INTERCONVERSION

*Proof.* In this section, we establish the superiority of our augmented bipartite graph framework against existing linear spatial-temporal transformation architecture by demonstrating that conventional linear spatial-temporal interconversion mechanisms and spatial interaction layers emerge as degenerated case of our general formulation.

Specifically, the M-RNN architecture and its SOTA derivatives (GRIN, GSLI, Imputeformer) employ interleaved spatial-temporal operations that correspond to a restricted parameterization of our signal component customization in Section 4.3.2 or noise component customization in Section 4.3.2. These constraints manifest in two primary forms:

1. **Structural Degeneracy**: By enforcing $\mathbf{E}^{\mathrm{S}}\mathbf{W}^{(l)}\mathbf{E}^{\mathrm{S},\top} = \mathbf{A}$ (where $\mathbf{A}$ denotes the given adjacency matrix from $\mathcal{G}^{\mathrm{S}}$), existing methods collapse our learnable bipartite projections to static graph representations.

2. **Form Restriction**: The sparse spatial transformer in Imputeformer (Nie et al., 2024) corresponds to a constrained instantiation of our propagation function for signal, where adaptive edge weights $\mathbf{W}^{(l)}$ are replaced by isolated attention matrices $\{\mathbf{Q}_h\mathbf{K}_h^\top\}_{h=1}^H$ with predefined sparsity patterns, where $H$ denotes the number of attention heads.

The architectural degeneracy inherent in existing designs imposes strict constraints on their theoretical expressiveness, which mathematically guarantees inferior performance relative to our augmented bipartite graph framework under any learnable parameterization. $\qquad\square$

## A.3 VALIDITY OF DGP ASSUMPTIONS

We adopt the assumption that the data-generating process (DGP) takes the form $\mathbf{X}_{i,\tau} = f(\mathbf{X}_{i,<\tau}) + \varepsilon_{i,\tau}$, where we refer to $f(\mathbf{X}_{:,<\tau})$ as the signal component, and to $\varepsilon_{i,\tau}$ as the noise component. We further impose three assumptions on the statistical properties of $\varepsilon_{i,\tau}$. Among these assumptions, the second assumption $\varepsilon_{i,\tau} \perp \varepsilon_{j,\omega}, \forall j \in \{1,\ldots,N\}, \omega \neq \tau$ entails that the noise component lacks temporal dependency, may appear overly restrictive or insufficiently general. Therefore, in this section, we provide a mathematical justification to support the generality of this assumption.

Consider a DGP whose $\varepsilon_{i,t}$ does not satisfy the second assumption (i.e. it exhibits temporal dependencies). For clarity, we refer to its $f(\mathbf{X}_{i,<\tau})$ as the **trend** and to its $\varepsilon_{i,\tau}$ as the **shock**. Since shock exhibits temporal structures, it can be decomposed into a predictable component and an unpredictable remainder:

$$\varepsilon_{i,\tau} = g(\varepsilon_{:,<\tau}) + \nu_{i,\tau} \tag{17}$$

where $g(\cdot)$ is a non-linear autoregressive function describing the predictable expected exogenous shock component based on past **shock information**, and $\nu_{i,\tau}$ is the non-predictable component independent of **any previous information**. Moreover, since the exogenous shock $\varepsilon_{i,\tau} = \mathbf{X}_{i,\tau} - f(\mathbf{X}_{:,<\tau})$, it is itself a function of $\mathbf{X}_{:,\leq\tau}$. Hence, we can equivalently write:

$$\varepsilon_{i,\tau} = \tilde{g}(\mathbf{X}_{i,<\tau}) + \nu_{i,\tau} \tag{18}$$

where $\tilde{g}(\cdot)$ is a non-linear autoregressive function describing the predictable expected exogenous shock component based on past **observations**. Substituting this expression back to the original DGP yields

$$\mathbf{X}_{i,\tau} = f(\mathbf{X}_{i,<\tau}) + \tilde{g}(\mathbf{X}_{i,<\tau}) + \nu_{i,\tau} \tag{19}$$

since both $f(\cdot)$ and $\tilde{g}(\cdot)$ takes past observations as inputs, we can combine them into a single autoregressive function $h(\cdot)$:

$$\mathbf{X}_{i,\tau} = h(\mathbf{X}_{i,<\tau}) + \nu_{i,\tau} \tag{20}$$

where $h(\mathbf{X}_{i,<\tau}) := f(\mathbf{X}_{i,<\tau}) + \tilde{g}(\mathbf{X}_{i,<\tau})$; and $\nu_{i,\tau}$ is the previously defined non-predictable component independent from any previous information. We then define $h(\mathbf{X}_{i,<\tau})$ as the new **signal** and $\nu_{i,\tau}$ as the new **noise**. Under this reparameterization, the noise is, by construction, temporally independent of the past, thereby satisfying the three assumptions imposed on noise.

To further demonstrate and clarify these claims, we've also examined components decomposed by the model and compared against the original trend and shock, as well as DGP-defined signal and noise in Section D.2.

## B  IMPLEMENTATION DETAIL

### B.1  DATASETS

**Air Quality Datasets**   **AQI** contains hourly PM2.5 measurements collected from 437 monitoring stations in Beijing from 2014/05/01 01:00:00 to 2015/04/30 23:00:00. We adopt the standard evaluation split following Yi et al. (2016); Cao et al. (2018), where data from March, June, September, and December is used for testing, and the remaining for training. Adjacency matrix for the exogenous spatial graph are constructed using geographical coordinates of each sensor as in Yi et al. (2016); Cao et al. (2018). And **AQI36** is a subset including the first 36 monitoring stations.

| Name | $N$ | $T$ | $C$ | Interval | Type |
|------|-----|-----|-----|----------|------|
| AQI | 437 | 8760 | 1 | 1h | Air Quality |
| AQI36 | 36 | 8759 | 1 | 1h | Air Quality |
| METR-LA | 207 | 34272 | 1 | 5min | Traffic |
| PEMS-BAY | 325 | 52128 | 1 | 5min | Traffic |
| PEMS03 | 358 | 26208 | 1 | 5min | Traffic |
| PEMS04 | 307 | 16992 | 3 | 5min | Traffic |
| PEMS07 | 883 | 28224 | 1 | 5min | Traffic |
| PEMS08 | 170 | 17856 | 3 | 5min | Traffic |
| HAR | 561 | 10299 | 1 | 1.28s | Healthcare |
| ETTm1 | 7 | 69680 | 1 | 15min | Smart Grid |
| ETTm2 | 7 | 69680 | 1 | 15min | Smart Grid |
| ELERGONE | 370 | 105215 | 1 | 15min | Smart Grid |

Table 2: Statistical Summary of Datasets

**Traffic Datasets**  We adopt two subtypes of traffic datasets for testing. **METR-LA** and **PEMS-BAY** provide road speed data sampled every 5 minutes from sensors in Los Angeles and San Francisco Bay areas. **PEMS03**, **PEMS04**, **PEMS07**, and **PEMS08** consist of traffic volume data collected by Cal-trans Performance Measurement System (PeMS), also sampled at 5-minute intervals. For these, we construct adjacency matrix for the exogenous spatial graph according to the official highway networks.

**Human Activity Recognition Dataset**  **HAR** (Human Activity Recognition) is a popular dataset consisting of sensor signals collected from wearable devices worn by volunteers as they perform various physical activities. The data are pre-processed to a sampling interval of 1.28 seconds and comprise 561 different sensors, including accelerometers and gyroscopes from different body locations. As the dataset does not provide explicit exogenous spatial graph, we construct a fully connected graph among all sensors to allow comprehensive modeling of sensor interactions. In addition to evaluating imputation error, **HAR** allows us to evaluate the imputation quality on downstream classification tasks, using activity labels provided for each sequence.

**Smart Grid Dataset**  We adopt two subtypes of smart grid dataset for testing. **ETTm1** and **ETTm2** are two Electric Transformer Temperature dataset each comprising 7 sensor measurements, including load and oil temperature from an electrical transformer station. These datasets cover the period from 2016/07 to 2018/07 with a 15-minute sampling interval. **ELERGONE** contains electricity consumption records from 370 clients, sampled every 15 minutes from 2011 to 2014. For all these datasets, we follow the common data splitting protocol, allocating 10% of the data for validation and 20% for testing. As no exogenous spatial graphs are provided for these datasets, we adopt a fully connected adjacency matrix for models that require graph information.

### B.2 EVALUATION METRICS AND DATASET MASKING STRATEGY

We use both the RMSE and MAE for evaluation. The definition of MAE is as follows:

$$\text{MAE} = \frac{\sum_{i=1}^{N} \sum_{\tau=1}^{W} \mathbf{M}_{i,\tau} \cdot \left| \widehat{\mathbf{X}}_{i,\tau} - \mathbf{X}_{i,\tau} \right|}{\sum_{i=1}^{N} \sum_{\tau=1}^{W} \mathbf{M}_{i,\tau}} \tag{21}$$

The definition of RMSE is as follows:

$$\text{RMSE} = \sqrt{\frac{\sum_{i=1}^{N} \sum_{\tau=1}^{W} \mathbf{M}_{i,\tau} \cdot \left( \widehat{\mathbf{X}}_{i,\tau} - \mathbf{X}_{i,\tau} \right)^2}{\sum_{i=1}^{N} \sum_{\tau=1}^{W} \mathbf{M}_{i,\tau}}} \tag{22}$$

To ensure experimental consistency and fair benchmarking, we adopt data masking strategies similar to previous work (Nie et al., 2024; Cini et al., 2021). For **AQI** and **AQI36**, simulated sensor faults are generated following the original procedure in ST-MLV (Yi et al., 2016), where an observation is manually dropped if it isn't naturally missing and the observation from the same time last month is missing. For all other datasets, we evaluate both point missing and block missing scenarios, as commonly employed in (Cao et al., 2018; Cini et al., 2021; Nie et al., 2024):

1) Point missing: Missing values are sampled independently for each observation using a Bernoulli distribution $\mathbf{M}_{i,\tau} \sim \mathrm{B}\left(1, 0.25\right)$, where each observation is randomly masked with a probability of 25%. 2) Block missing: This scenario simulates more realistic sensor failures with continuous outages. First an initial mask $\mathbf{M}_1$ is generated using the same point-wise Bernoulli process as above but with a lower probability of 5%. Next, to mimic outages, a start mask $\mathbf{M}_2$ is created by sampling $\mathbf{M}_{2,i,\tau} \sim \mathrm{B}\left(1, 0.0015\right)$ for each sensor $i$ and time step $\tau$. Each activated failure represents a continuous outage of length $\mathbf{L}_{i,\tau} \sim \mathrm{Uniform}\left(12, 48\right)$. The final block missing mask is computed as the following equation:

$$\mathbf{M} = \mathbf{M}_1 \vee \mathrm{Span}\left(\mathbf{M}_2, \mathbf{L}\right) \tag{23}$$

where $\vee$ denotes the logical OR operation and $\mathrm{Span}\left(\cdot, \cdot\right)$ expands each selected outage starting point into a continuous missing block of the specified length. Since we train all our model under seed = 2, all models share the same missing mask for each dataset under the same missing pattern.

For **AQI** and **AQI36**, we follow the original procedure in ST-MLV (Yi et al., 2016) and uses March, June, September and December as the testing set; 10% of the remaining adjacent but non-overlapping data-points preceding each testing month is used as the validation set; and the remaining non-overlapping data-points are used as the training set. For all other datasets, we follow previous works (Nie et al., 2024; Cini et al., 2021) and uses the beginning 70% data-points for training; the next non-overlapping 10% for validation and the final non-overlapping 20% for testing. The non-overlapping segmentation guarantees that none of the validation and testing set will be visible to the model during training.

During training, we randomly mask out an additional 10% if the non-missing observations to facilitate more rigid model training and avoid overfitting on specific missing patterns. See Section B.3 for the loss function used during training.

Following previous works Nie et al. (2024), for non-matrix factorization methods, we evaluate the model's out-of-sample (OOS) imputation performance, ensuring that no information from the validation or testing set is exposed to the model during training. For in-sample methods (i.e., TIDER, adaTIDER and LCR), we evaluate the mode's in-sample (IS) imputation performance, where only the ground-truth of dropped observations in testing and validation sets are not shown to the model during training.

## B.3  Train Loss Design

To enable effective self-supervised learning, we adopt a dual-objective training strategy following previous work Du et al. (2023). The loss function combines observed reconstruction loss (ORL) and masked imputation loss (MIL) to ensure accurate reconstruction of missing values. Specifically, the loss functions are defined as:

$$l_{\mathrm{MIL}} = \frac{\sum_{i=1}^{N} \sum_{\tau=1}^{W} \left(1 - \mathbf{M}_{i,\tau}^{(t)}\right) \cdot \ell\left(\widehat{\mathbf{X}}_{i,\tau}, \mathbf{X}_{i,\tau}^{(t)}\right)}{\sum_{i=1}^{N} \sum_{\tau=1}^{W} \left(1 - \mathbf{M}_{i,\tau}^{(t)}\right)} \tag{24}$$

$$l_{\mathrm{ORL}} = \frac{\sum_{i=1}^{N} \sum_{\tau=1}^{W} \mathbf{M}_{i,\tau}^{(t)} \cdot \ell\left(\widehat{\mathbf{X}}_{i,\tau}, \mathbf{X}_{i,\tau}^{(t)}\right)}{\sum_{i=1}^{N} \sum_{\tau=1}^{W} \mathbf{M}_{i,\tau}^{(t)}} \tag{25}$$

where $\ell(\cdot, \cdot)$ represents element-wise reconstruction loss (e.g. mean absolute error). The overall training loss combines both components:

$$l = l_{MIL} + l_{ORL} \tag{26}$$

GraphTSI is updated by minimizing the final loss $l$.

# C ADDITIONAL RESULTS

## C.1 BENCHMARK RMSE RESULTS

| Dataset | Mask | Avg. | GRIN | GSLI | BRITS | TIDER | adaTIDER | LCR | UnIMP | SAITS | Imputeformer | GraphTSI | |
|---|---|---|---|---|---|---|---|---|---|---|---|---|---|
| AQI36 | Simulated | 56.9312 | 22.0190 | 27.1960 | 23.1762 | 34.7269 | 77.0252 | 68.0342 | 34.5819 | 36.4735 | 27.1911 | 19.3724 | -12.020% |
| AQI | | 67.2950 | 27.5155 | 29.3724 | 33.8573 | 48.8622 | 48.6232 | 51.4744 | 35.0683 | 36.1205 | 29.9331 | 24.9411 | -9.356% |
| METR-LA | Point | 22.2447 | 3.7283 | 3.5297 | 8.3640 | 13.7551 | 14.3426 | 13.4495 | 4.4427 | 6.9162 | 3.5208 | 3.4367 | -2.389% |
| | Block | 22.3203 | 5.8599 | 8.9805 | 15.3985 | 13.5838 | 14.3068 | 7.2275 | 8.4585 | 6.0490 | 5.7516 | -1.848% |
| PEMS-BAY | Point | 9.3962 | 1.2245 | 1.1986 | 3.7387 | 6.4565 | 6.8333 | 7.5620 | 1.6796 | 3.0573 | 1.1591 | 1.0761 | -7.163% |
| | Block | 9.4969 | 2.4365 | 2.5574 | 3.7126 | 6.4292 | 6.7854 | 7.6067 | 3.2665 | 3.6806 | 2.4118 | 2.2382 | -7.198% |
| PEMS03 | Point | 110.4516 | 14.6384 | 13.0551 | 26.0660 | 49.5692 | 53.4239 | 52.2824 | 17.1393 | 28.1297 | 13.3883 | 11.5974 | -11.166% |
| | Block | 110.5333 | 20.2423 | 15.8434 | 23.1337 | 45.3507 | 56.4151 | 47.0515 | 24.8856 | 26.1246 | 16.0300 | 14.2961 | -9.766% |
| PEMS04 | Point | 74.6774 | 14.8118 | 8.8302 | 23.7165 | 34.4027 | 36.0054 | 36.2034 | 15.9019 | 21.6576 | 15.6002 | 8.6081 | -2.515% |
| | Block | 74.8785 | 23.0640 | 16.3150 | 22.9959 | 33.6004 | 34.3942 | 36.0808 | 19.0009 | 21.8728 | 17.7224 | 14.8050 | -9.255% |
| PEMS07 | Point | 149.5555 | 20.0317 | 17.9646 | 44.5825 | 71.5680 | 73.2541 | 66.7169 | 26.9325 | 47.6898 | 18.7524 | 17.8850 | -0.443% |
| | Block | 149.9398 | 27.9339 | 25.8423 | 45.0385 | 69.8824 | 73.2640 | 67.3023 | 34.4643 | 48.9659 | 27.9847 | 25.8727 | +0.118% |
| PEMS08 | Point | 65.4021 | 10.9649 | 6.3271 | 21.9158 | 30.3188 | 33.2092 | 28.8825 | 11.9393 | 17.8621 | 11.1520 | 5.9480 | -5.992% |
| | Block | 66.1944 | 19.9088 | 13.8394 | 20.4427 | 29.7691 | 17.3010 | 32.4780 | 28.8512 | 18.6814 | 13.1710 | 12.0533 | -8.486% |
| HAR | Point | 0.3237 | D.N.F. | 0.0719 | 0.1058 | 0.2341 | 0.2010 | 0.2469 | 0.1205 | 0.1105 | 0.0781 | 0.0679 | -5.550% |
| | Block | 0.3162 | D.N.F. | 0.0800 | 0.0946 | 0.1949 | 0.1321 | 0.1973 | 0.2414 | 0.1061 | 0.0752 | 0.0668 | -11.260% |
| ETTm1 | Point | 5.9672 | 0.7907 | 0.4685 | 0.5749 | 4.3586 | 3.6917 | 5.4398 | D.N.F. | 0.4805 | 0.4779 | 0.4469 | -4.610% |
| | Block | 6.0152 | D.N.F. | 1.1718 | 1.4275 | 4.3706 | 3.7317 | 5.5418 | D.N.F. | 0.9531 | 1.3146 | 1.0806 | +13.373% |
| ETTm2 | Point | 8.4321 | D.N.F. | 0.5602 | 1.1929 | 5.3972 | 4.4453 | 7.2473 | D.N.F. | 0.6114 | 0.6067 | 0.4892 | -12.674% |
| | Block | 8.2934 | D.N.F. | 1.5953 | 2.3899 | 5.3795 | 4.3477 | 7.0941 | D.N.F. | 1.3585 | 2.0353 | 1.2853 | -5.388% |
| Elergone | Point | 2244.2345 | D.N.F. | 305.0960 | 628.0602 | 1467.8114 | 1557.9351 | 1540.4912 | D.N.F. | 567.2914 | 320.9664 | 335.6993 | +10.031% |
| | Block | 2162.3831 | D.N.F. | 489.3702 | 613.5619 | 1345.2236 | 1431.5295 | 1378.5088 | D.N.F. | 522.3876 | 413.4545 | 408.0314 | -1.312% |
| *Average Improvement* | | -82.671% | -21.821% | -8.915% | -43.408% | -70.273% | -72.066% | -73.064% | -34.896% | -37.725% | -14.609% | — | |

Table 3: RMSE Performance of models. The final column shows average performance gain over each benchmark setup, and the final row shows average performance gain over each model. D.N.F. indicates not finished within 8 hours for ETTm1, ETTm2 and Elergone datasets and not finished within 48 hours for other datasets.

In this experiment, we report the imputation performance of different methods on multivariate time series data using RMSE. The results are summarized in Table 3, covering 9 datasets and different masking strategies. As shown in the table, transformer-based approaches such as Imputerformer and graph-based method GSLI, achieve competitive performance on the evaluated datasets. Moreover, our proposed method, GraphTSI, consistently achieves the best or second-best RMSE across almost all datasets. Notably, GraphTSI achieves an average RMSE improvement of 15.260% over Imputerformer and 9.064% over GSLI. These results highlight the effectiveness of GraphTSI for accurate time series imputation.

## C.2 ADDITIONAL RESULTS FOR MISSING RATES

To further examine the robustness of different methods, we evaluate imputation performance on METR-LA and PEMS-BAY under different missing rates. We include Imputeformer and GSLI as baseline since their outstanding performance in previous experiments. The results are shown in Figure 7. On both METR-LA and PEMS-BAY datasets, the MAE values increase steadily as the missing rate rises from 25% to 95%. Overall, GraphTSI consistently achieves the lowest error across all missing rates, demonstrating strong robustness even in extremely sparse conditions.

## C.3 ADDITIONAL RESULTS FOR WINDOW SIZES

To further evaluate the impact of window sizes on different methods, we investigate imputation performance on METR-LA and PEMS-BAY under varying window sizes. We include Imputeformer and GSLI as baselines due to their competitive performance in previous experiments. The results are presented in Figure 8. On both datasets, increasing the window size generally leads to improved accuracy for all methods. However, for extremely long window sizes ($W > 36$), the increasing window size provide marginal improvement, and performance slowly degrades due to larger variances created by smaller batch sizes. Notably, GraphTSI consistently achieves the lowest RMSE across all window sizes, highlighting its stability and effectiveness in capturing temporal dependencies under different temporal contexts.

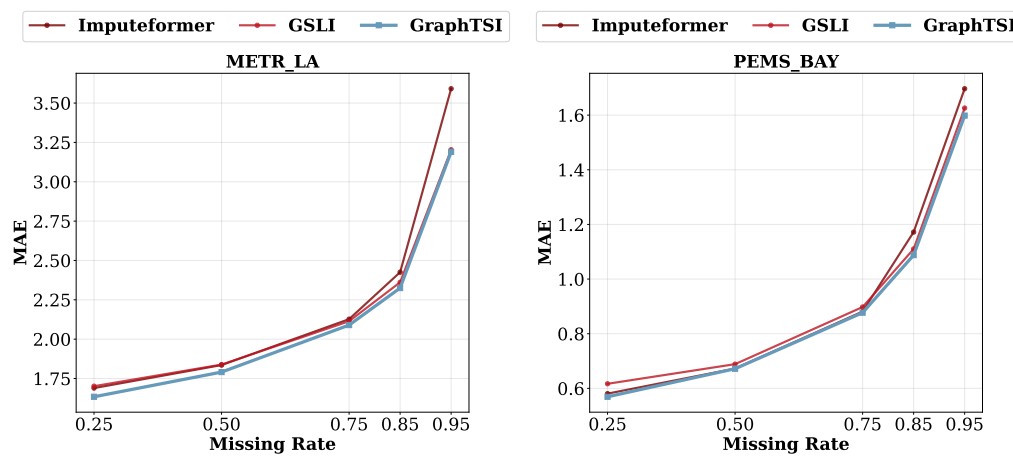

Figure 7: Results of different missing rate on METR-LA and PEMS-BAY datasets.

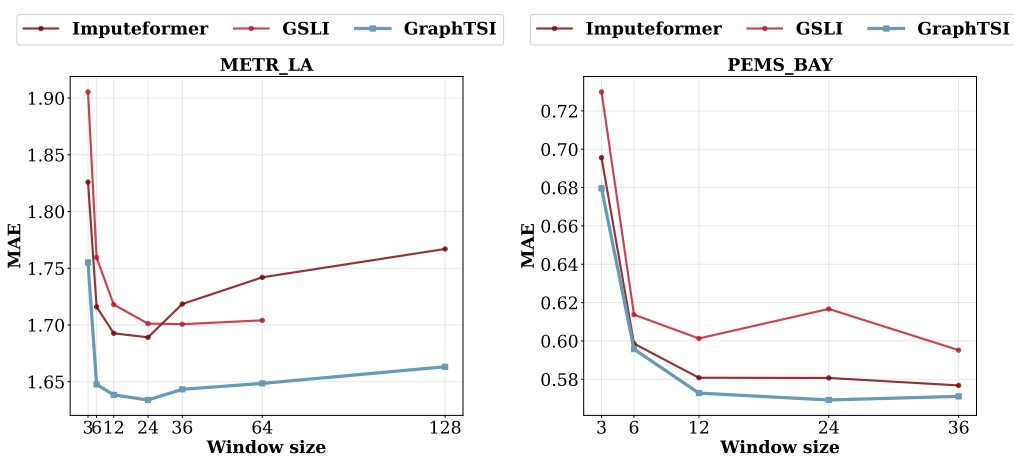

Figure 8: Result of different window sizes on METR-LA and PEMS-BAY datasets.

## C.4 ADDITIONAL RESULTS FOR DIFFERENT MISSING PATTERNS

In this section, we examine model performance under extreme missing patterns in PEMS08 dataset. We include Imputeformer and GSLI as baselines due to their competitive performance in previous experiments. Following prior works (Khayati et al., 2020), we tested model performance on the following missing patterns: 1) **Overlap**: Block missing overlaps with each other, meaning at least one sensor fails at every time step; 2) **Blackout**: Block missing happens unanimously across all sensors; 3) **Sensor Failure**: One of the sensors fails throughout the entire test set. The results are presented in Table 4. These results demonstrated that GraphTSI consistently outperforms strong baselines across all extreme missingness scenarios. Notably, under Blackout—when all sensors are missing concurrently, GraphTSI reduces MAE by 53.44% relative to the best baseline, indicating robust cross-sensor and temporal interpolation. In Overlap and Sensor Failure settings, GraphT-

| Missing Pattern | GSLI | Imputeformer | GraphTSI | |
|---|---|---|---|---|
| Overlap | 7.7746 | 6.8932 | 6.6936 | -2.8956% |
| Blackout | 38.1689 | 29.1509 | 13.5714 | -53.4443% |
| Sensor Failure | 10.6389 | 10.1796 | 9.3771 | -7.8834% |

Table 4: MAE performance of models under different missing patterns.

SIachieves 2.90% and 7.88% relative MAE reductions, respectively, suggesting that the method remains effective when missing blocks are pervasive or when a single sensor is entirely absent.

# D CASE STUDY

## D.1 TEMPORAL ATTENTION FOR LONG WINDOW

In this section, we demonstrate the attention matrix of the forward bi-unidirectional predictor to demonstrate the locality of attention spans in imputation tasks. Results are shown in Figure 9.

Of these attention heads, we can identify three primary patterns: 1) the third head in the first layer aggregates information from the beginning of each series; 2) the second head in the first layer and the fourth head in the third layer exhibit a pronounced periodic structure that decays gradually with increasing lag; 3) the remaining heads display strong locality, focusing predominantly on a narrow temporal window preceding each time step.

These results help explain the marginal gains in imputation accuracy observed for extremely long window sizes. They also highlight opportunities for future work through discrete transformers that reduce computation time with minimal impact on performance.

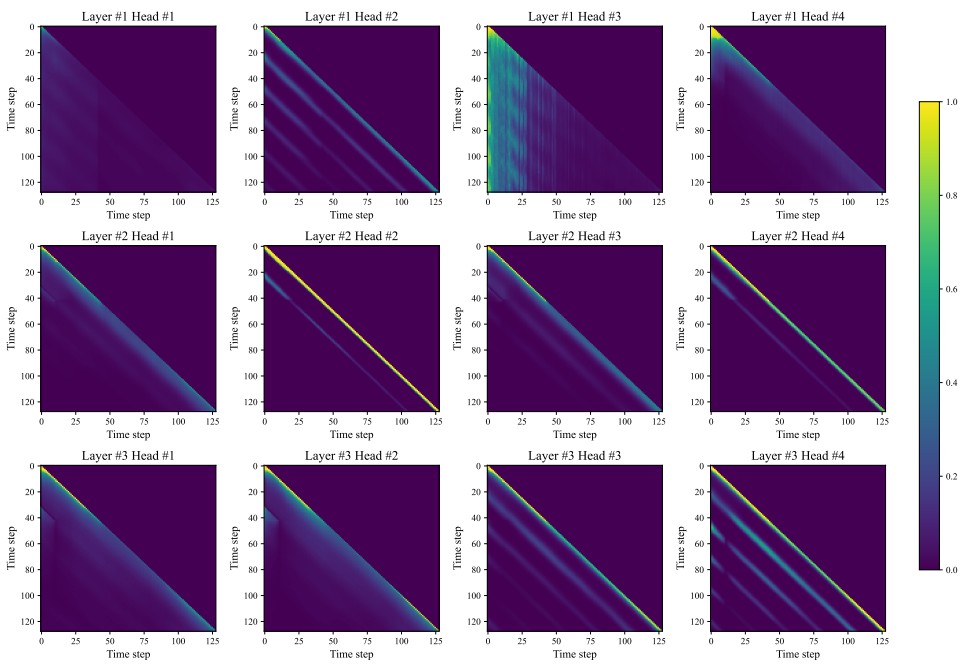

Figure 9: Attention matrices across all heads and layers of GraphTSI, evaluated on the METR-LA dataset. Each entry is aggregated as the 99th percentile over the batch and sensor dimension.

## D.2 SEPARATION OF SIGNAL AND NOISE

In this section, we examine the behavior of our model when exogenous shock exhibit temporal autocorrelation, as described in Section A.3, using an artificially generated dataset. First, we will establish the experiment setup; Next, we calculate the theoretical value of **trend** and **shock** based on the setup, as well as the theoretical value of **signal** and **noise** based on the DGP; Finally, we demonstrate the actual component decomposed and imputed by our model and compare them against each of the two pairs to provide empirical proof for the validity of our assumptions and the explainability of our model.

To construct the artificial dataset, we set the number of channels to $C = 1$ and construct each measurement from series $i$ and time step $\tau$ as follows:

$$\mathbf{X}_{i,\tau} := \sin\left(\phi_i + \omega\tau\right) + \varepsilon_{i,\tau} \tag{27}$$

where $\phi_i$ is the initial phase of the sinusoidal trend; $\omega$ is the frequency; $\sin\left(\phi_i + \omega\tau\right)$ represents the entire sinusoidal trend; and $\varepsilon_{i,\tau}$ represents the exogenous shock defined as a Moving Average process:

$$\varepsilon_{i,\tau} = \sum_{\Delta=0}^{l-1} \nu_{i,\tau-\Delta} \tag{28}$$

where $l$ is the lag of this $\mathrm{MA}(l)$ process, and $\nu_{i,\tau}$ is the underlying multivariate normally-distributed noise with zero mean and covariance matrix $\Sigma$:

$$\nu_{:,\tau} \sim \mathrm{N}\left(\mathbf{0}, \Sigma\right) \tag{29}$$

Under this setup, the sinusoidal **trend** is defined as $\sin\left(\phi_i + \omega\tau\right)$, and the exogenous **shock** is defined as $\varepsilon_{i,t}$. However, according to our DGP, the signal component should describe the predictable expected observation based on past information. Here, the sinusoidal trend via averaging through the past; the exogenous shock is partially predictable through a linear combination of $\varepsilon_{:,t-1}$, and $\varepsilon_{\neq i,t}$. Therefore, by definition, the target of **signal** should be $\mathbf{X}_{i,\tau} - \nu_{i,\tau}$, and the target of **noise** should be $\nu_{i,\tau}$. To test these claims, we build datasets with $N = 3, W = 24, C = 1, l = 9, \phi_i = 2i\pi/3, \omega = 2\pi/96$ and the covariance matrix as

$$\Sigma = \begin{pmatrix} 1 & 0.5 & 0.5 \\ 0.5 & 1 & 0.5 \\ 0.5 & 0.5 & 1 \end{pmatrix}$$

We employ the standard point-dropping mechanism to train and evaluate our model. The output hidden states of the bi-unidirectional predictors from both sides are passed through the MLP readout to retrieve our decomposed signal component, and the noise component is defined as the difference between our final imputation and the decomposed signal. Results are shown in Figure 10 and Figure 11. As we can clearly see, our decomposed signal component resembles more to the ground truth signal target, which is defined as $\mathbf{X}_{i,\tau} - \nu_{i,\tau}$, and the noise component resembles more to the one-period cross-sectional noise $\nu_{i,\tau}$ rather than the temporally correlated shock.

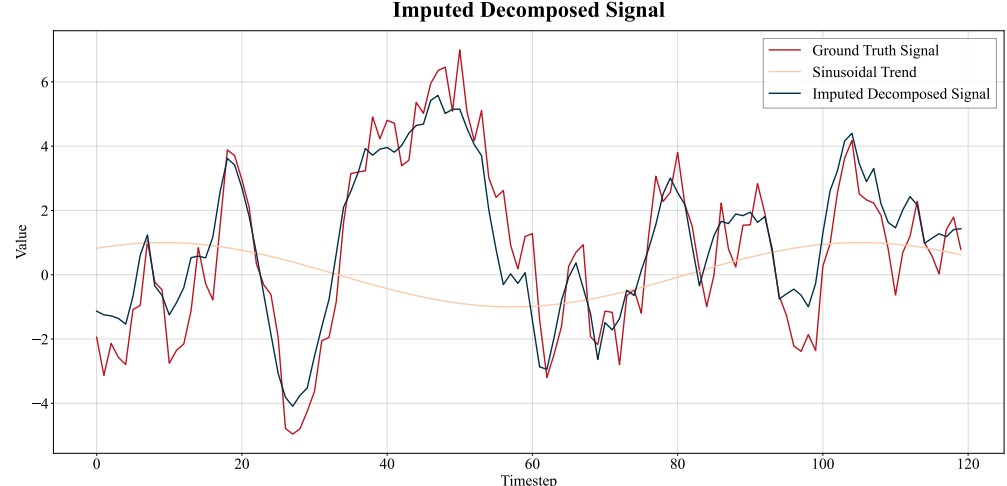

Figure 10: Decomposed signal component compared against sinusoidal trend and DGP-defined signal component

## LLM USAGE

To enhance clarity and readability, we use Large Language Models (LLMs) as a general-purpose writing assistant for polishing. Specifically, after drafting the manuscript ourselves, we use LLMs

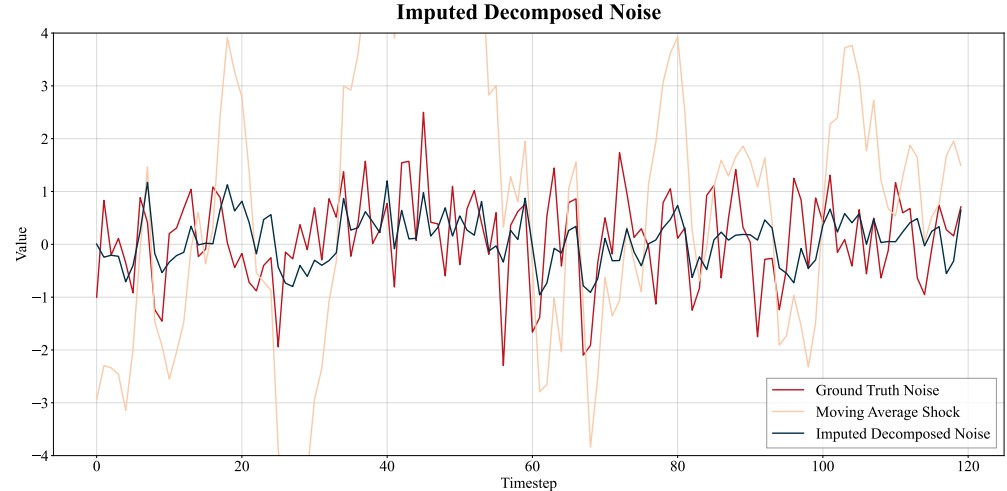

Figure 11: Decomposed noise component compared against exogenous moving average shock and DGP-defined noise component

to: 1) Suggest alternative phrasing to avoid repetitiveness while being consistent; 2) Check for consistency in tense, notation and other grammatical issues; 3) Asking for more standard terminology or abbreviations. We have reviewed and edited all LLM-generated text to ensure accuracy and faithfulness, and no text was incorporated without verification from authors.

The LLMs are NOT used for research ideation, experiment design or data analysis. All technical claims, equations, proofs, and conclusions are written and verified by the authors. We take responsibility for the content of this paper.

