# OpenReview forum: "Multivariate Time Series Imputation with Signal-Noise Disentangled Graph Propagation"
_ICLR.cc/2026/Conference — Submitted to ICLR 2026_

### Official Review · Reviewer_6ZuJ · 2025-10-15

**Soundness:** 3
**Presentation:** 3
**Contribution:** 2
**Rating:** 4
**Confidence:** 3

**Summary:**

This paper proposes GraphTSI, a graph-based model for multivariate time series imputation that explicitly separates each sequence into signal and noise components. To address the limitations of insufficient component separation and weak spatial–temporal modeling in existing models, GraphTSI introduces a prediction–subtraction mechanism and an augmented bipartite graph for adaptive information exchange between spatial and temporal representations.
Across nine real-world datasets, GraphTSI achieves the best performance, reducing MAE by 17.2% over Imputeformer and 9.6% over GSLI. Additional experiments confirm its robustness, interpretability, and strong performance even with up to 95% missing data.

**Strengths:**

This paper makes a good contribution to multivariate time series imputation. The proposed GraphTSI model is  novel and interesting, combining temporal prediction and spatial reasoning in a unified, interpretable architecture. The introduction of signal–noise decomposition is  insightful.
Experimentally, the paper is comprehensive and convincing. Results across nine diverse real-world datasets demonstrate consistent and significant improvements over strong baselines. The ablation studies, robustness tests, and downstream task evaluations provide thorough evidence supporting the model’s design choices.
Finally, the paper is well-structured and clear.

**Weaknesses:**

W1: The paper does not clearly explain the backward propagation mechanism in the bi-unidirectional design.

W2: The computational complexity of GraphTSI is not adequately analyzed.

W3: The dataset selection is heavily skewed toward traffic domains. It would strengthen the generality of the model to include datasets from other domains such as healthcare, energy, or finance.

W4: The set of baseline models could be expanded.

[1] Zhang S, Wang S, Miao H, et al. Score-cdm: Score-weighted convolutional diffusion model for multivariate time series imputation. IJCAI 2024

[2] Liang G, Tiwari P, Nowaczyk S, et al. Higher-order spatio-temporal physics-incorporated graph neural network for multivariate time series imputation. CIKM 2024

[3] Gao S, Koker T, Queen O, et al. Units: A unified multi-task time series model. Neurips 2024

[4] Ahmed N, Yalavarthi V K, Schmidt-Thieme L. Motif-aware Graph Neural Networks for Networked Time Series Imputation. AAAI 2025

[5] Liu S, Li X, Chen Y, et al. Disentangling Dynamics: Advanced, Scalable and Explainable Imputation for Multivariate Time Series. TKDE 2025

[6]  Chen X,  Cheng Z, Cai H, Saunier N, Sun L: Laplacian Convolutional Representation for Traffic Time Series Imputation. TKDE 2024

W5: It would be helpful to evaluate the model under diverse missing patterns (disjoint, MCAR, Overlap, Blackout)

[7] Mind the Gap: An Experimental Evaluation of Imputation of Missing Values Techniques in Time Series

**Questions:**

Q1: Using the reversed time series for imputation (backward) might raise concerns about potential information leakage. Does leveraging future patterns to enhance the semantic representation of current time steps constitute a form of “cheating”?

Q2: In the augmented bipartite graph, the number of spatial and temporal nodes usually differs. How does the model ensure stable message passing in extreme cases, such as very long sequences or highly sparse graphs?

Q3: The signal–noise decomposition assumes independent noise components, but in many real settings, noise can be temporally autocorrelated or cross-variable correlated. Would the proposed prediction–subtraction mechanism remain stable and effective in such cases?

I will reconsider my rating if the questions and weakness are handled properly.

---

> ### Author Response · Authors · 2025-11-24
> **Response to Reviewer 6ZuJ [Part 1]**
>
> Thank you for your contentful review. Below are our response to your concerns and questions:
>
> ---
>
> **1. Regarding implementation detail**
>
> > W1: The paper does not clearly explain the backward propagation mechanism in the bi-unidirectional design.
>
> Thank you for your comment. Our bi-unidirectional design consists of two independent unidirectional predictors that are trained in parallel (logically), similar to a Siamese setup. After the missing-aware input embedding, both predictors take the same input representation $h\_i^{(l-1)}$. Each predictor independently produces a univariate signal feature $\\tilde{h}\_i^{(l),\\mathrm{signal},\\mathrm{f}}$, which is then passed through the augmented bipartite graph to retrieve its own multivariate signal feature. An MLP readout maps each multivariate feature to an imputed univariate noise estimate. Finally, the two directional estimates are merged to form the complete noise representation $\\tilde{h}\_{i,\\tau}^{(l),\\mathrm{noise}}$. The execution of the two predictors is order-agnostic; in our code, we compute the backward-direction branch first for intuitiveness in code design, but **the branches are logically parallel and non-interacting** in the forward pass.
>
> Backpropagation strictly follows the computational graph and the chain rule. The gradient of the loss with respect to $\\tilde{h}\_{i,\\tau}^{(l),\\mathrm{noise}}$ is distributed to the two directional outputs, which backpropagate independently along their respective branches. At the shared input $h\_i^{(l-1)}$, **the gradients from both branches are summed**, yielding the final gradient for $h\_i^{(l-1)}$. This process continues utill it reaches the raw input $\\mathbf{X}^{(t)}$, and all learnable parameters are updated according to their accumulated gradients.
>
> We hope this explanation addresses your questions.
>
> > W2: The computational complexity of GraphTSI is not adequately analyzed.
>
> In general, our model consists of three main parts: the missing-aware input embedding, the bi-unidirectional predictors, and the augmented bipartite graph. Specifically, given a windowed series of size $W\\times S\\times C$, where $W$ is the window size, $S$ is the number of series, and $C$ is the number of channels per series:
>
> - **missing-aware input embedding:** $O(WNCd\_{\\mathrm{h}})$, where $d\_{\\mathrm{h}}$ is the size of hidden states;
> - **bi-unidirectional predictors:** $O(W^2Nd\_{\\mathrm{h}})$, since attention scales quadratically with length
> - **augmented bipartite graph:** gather and diapatch of signal component is $O(WHNd\_{\\mathrm{h}})$; gather and dispatch of noise component is $O(WNd\_{\\mathrm{h}})$; propagation of signal component is $O(\\left|\\mathcal{E}^{\\mathrm{S}}\\right|Hd\_{\\mathrm{h}})$; and propagation of noise component is $O(\\left|\\mathcal{E}^{\\mathrm{S}}\\right|Nd\_{\\mathrm{h}})$, where $H$ is the number of latent heads, and $\\left|\\mathcal{E}^{\\mathrm{S}}\\right|$ is the number of edges in exgenous spatial graph.
>
> Together, the model has complexity $O\\left(WNd\_{\\mathrm{h}}\\left(W+H+C\\right)+\\left|\\mathcal{E}^{\\mathrm{S}}\\right|d\_{\\mathrm{h}}(H+N)\\right)$, which scales quadratically with window sizes.
>
> In response to your suggestions, we've included an analysis of model complexity in Section 4.4 of our $\\underline{\\text{revised paper}}$. Furthermore, in Section 5.2 of our $\\underline{\\text{revised paper}}$, we've included a new experiment regarding model efficiency against other methods. Figure 6 of our $\\underline{\\text{revised paper}}$ demonstrated model speed for inference over relative imputation error. From these observations, we can see that GraphTSI delivers lower imputation error while performing just as fast as other baseline models.
>
> We hope this addresses your question.

---

> ### Author Response · Authors · 2025-11-24
> **Response to Reviewer 6ZuJ [Part 2]**
>
> > Q1: Using the reversed time series for imputation (backward) might raise concerns about potential information leakage. Does leveraging future patterns to enhance the semantic representation of current time steps constitute a form of “cheating”?
>
> Thank you for your questions. **The use of future patterns does NOT cause information leakage and is very crucial for multivariate time series imputation task.** Different from forecasting tasks that predict the future values, the imputation task utilizes the full context, both past and future, within a sliding window to recover missing values. Imagine a datapoint where all the sensors malfunction at the first timestep. The only way these datapoints can be accurately imputed is by using future information and imputing in reverse. In addition, most of the baseline models leverage bidirectional structures. For example, both BRITS and GRIN use Bi-LSTM-like or Bi-GRU-like structures in their model design; Imputeformer and SAITS use standard transformer structures to combine information from both sides. **Therefore, the use of reversed time series in imputation is crucial, especially for imputing missing points at the beginning of each window.**
>
> We hope this additional information addresses your concerns.
>
> > Q2: In the augmented bipartite graph, the number of spatial and temporal nodes usually differs. How does the model ensure stable message passing in extreme cases, such as very long sequences or highly sparse graphs?
>
> Thank you very much for your insightful question. We train separate models for each dataset and evaluate them separately, similar to previous works. Therefore, **given a specific dataset, the number of spatial nodes is the same as the number of sensors, and should stay fixed for a given model.**
>
> However, the number of temporal nodes, which is the same as the window size, and the bipartite edges, which only connect non-missing temporal-spatial pairs, can indeed change according to the given input. To ensure robust and stable message passing, **we employ a size-agnostic design** for the gather and dispatch functions of both signal and noise components. Specifically, we employ the projected latent attention for signal and simple concatenation for noise. These designs guarantee that the model transforms between spatial and temporal representations reliably and independently of window size or missing rates.
>
> To validate these claims, we have included additional experiments for model performance under long window sizes in Appendix C.2, and additional experiments for model performance under extreme missing rates in Appendix C.3 in our $\\underline{\\text{revised paper}}$.
>
> - **For long window sizes:** GraphTSI maintains stable imputation performance as the window size increases, with only marginal relative degradation due to the batch-size reduction at larger windows. At $W=64$, GraphTSI outperforms the transformer-based SOTA model Imputeformer by 5.37% on METR-LA and 54.94% on PEMS08, and exceeds the graph-based SOTA model GSLI by 3.26% on METR-LA and 6.6663% on PEMS08. At $W=128$, GraphTSI consistently surpasses Imputeformer by 5.87% on PEMS08 and 56.04% on METR-LA, while GSLI failed to complete training within 8 hours.
>
>     | window size | PEMS08 MAE | METR_LA MAE |
>     | :---------: | :--------: | :---------: |
>     |      3      |   2.0916   |   1.7551    |
>     |      6      |   1.8895   |   1.6476    |
>     |     12      |   1.8357   |   1.6384    |
>     |     24      |   1.8007   |   1.6339    |
>     |     36      |   1.8079   |   1.6433    |
>     |     64      |   1.8257   |   1.6485    |
>     |     128     |   1.8573   |   1.6632    |
>
> - **Extremely high missing rates:** As the missing rate increases, GraphTSI’s imputation error rises gradually, yet it consistently outperforms both SOTA baselines. Even at 95% missing rate, it delivers improvements of 1.70% over GSLI and 5.80% over Imputeformer.
>
> These results demonstrated the stability of GraphTSI even under extreme conditions. We hope this additional information addresses your concerns.

---

> ### Author Response · Authors · 2025-11-24
> **Response to Reviewer 6ZuJ [Part 3]**
>
> **2. Regarding the made assumptions**
>
> > Q3: The signal–noise decomposition assumes independent noise components, but in many real settings, noise can be temporally autocorrelated or cross-variable correlated. Would the proposed prediction–subtraction mechanism remain stable and effective in such cases?
>
> Thank you so much for your valuable question. However, we would like to clarify that **our assumptions allow noise to be cross-variable correlated, and we only make the assumption that noise is not temporally correlated.** While we acknowledge the fact that real-world *noise* collected can exhibit temporal autocorrelation, **the signal and noise are names we've attributed to the components our Data Generating Process and its assumptions create, and may differ from the general definition of these terms in what they represent.** We provide a simplified proof to validate the assumptions below:
>
> Suppose the noise does exhibit temporal correlation (To avoid confusion, hereafter we refer to temporally correlated noise as **shock**, and the other component as **trend**). We can further decompose it into what is predictable from past information using its autocorrelation property and the unpredictable remainder. Since predicting this shock uses only past information, we can incorporate this part with the **trend**, which we simply refer to as the **signal**, and the remainder as the **noise**. Now, the noise is temporarily uncorrelated, and we've transformed the original components (**trend** and **shock**) to DGP-compliant components (**signal** and **noise**), **guaranteeing the generality of our assumptions.** We've included a more detailed mathematical proof in Appendix A.3 of our $\\underline{\\text{revised paper}}$.
>
> For your convenience, we've added an additional case study in Appendix D.2 of our $\\underline{\\text{revised paper}}$, comparing the signal and noise components decomposed by the model against the theoretical value of **trend** and **shock**, as well as **signal** and **noise** using an artificially generated dataset. **The results provide empirical evidence that the model decomposes the original series into explainable components compliant with our assumptions.**
>
> We hope this additional information addresses your questions and concerns.
>
> **3. Regarding adding new datasets and baselines**
>
> > W3: The dataset selection is heavily skewed toward traffic domains. It would strengthen the generality of the model to include datasets from other domains such as healthcare, energy, or finance.
>
> Thank you for your comment. In our original experiments, we include datasets from three domains, including (1) Air quality (AQI, AQI36); (2) Healthcare (HAR); (3) Traffic (METR-LA, PEMS-BAY, PEMS03, PEMS04, PEMS07, PEMS08)
>
> To further enhance the diversity of domains, we've added three new datasets: ETTm1, ETTm2, and Elergone, and tested all baselines on them. Of these three datasets, ETTm1 and ETTm2 are electric transformer temperature datasets sampling measurements like electric load or oil temperature every 15 minutes. Elergone is a power consumption dataset recorded every 15 minutes regarding the power consumption of 370 clients. Here, we provide a summary of model performance against SOTAs on these datasets:
>
> | Dataset + Missing Pattern |  Imputeformer  |      GSLI      |     SAITS     | GraphTSI (ours) | Improvement |
> | :-----------------------: | :------------: | :------------: | :---------: | :-------------: | :---------: |
> |    ETTm1 Point Missing    |     0.4779     | $\\underline{0.4685}$  |    0.4805     |   **0.4469**    |   -4.610%   |
> |    ETTm1 Block Missing    |     0.5763     |     0.4991     | $\\underline{0.4483}$ |   **0.4179**    |   -6.781%   |
> |    ETTm2 Point Missing    |     0.3956     | $\\underline{0.3539}$  |    0.3869     |   **0.2752**    |  -22.238%   |
> |    ETTm2 Block Missing    |     1.2310     |     0.8787     | $\\underline{0.8460}$ |   **0.7001**    |  -17.246%   |
> |  Elergone Point Missing   |    27.7954     | $\\underline{27.3681}$ |    72.5428    |   **27.2700**   |   -0.358%   |
> |  Elergone Block Missing   | $\\underline{41.3560}$ |    44.7311     |    68.9966    |   **40.5752**   |   -1.888%   |
>
> We've updated Appendix B.1 regarding more detailed information on these datasets, and Table 1 regarding the MAE imputation results in our $\\underline{\\text{revised paper}}$.

---

> ### Author Response · Authors · 2025-11-24
> **Response to Reviewer 6ZuJ [Part 4]**
>
> > W4: The set of baseline models could be expanded.
> >
> > [1] Zhang S, Wang S, Miao H, et al. Score-cdm: Score-weighted convolutional diffusion model for multivariate time series imputation. IJCAI 2024
> >
> > [2] Liang G, Tiwari P, Nowaczyk S, et al. Higher-order spatio-temporal physics-incorporated graph neural network for multivariate time series imputation. CIKM 2024
> >
> > [3] Gao S, Koker T, Queen O, et al. Units: A unified multi-task time series model. Neurips 2024
> >
> > [4] Ahmed N, Yalavarthi V K, Schmidt-Thieme L. Motif-aware Graph Neural Networks for Networked Time Series Imputation. AAAI 2025
> >
> > [5] Liu S, Li X, Chen Y, et al. Disentangling Dynamics: Advanced, Scalable and Explainable Imputation for Multivariate Time Series. TKDE 2025
> >
> > [6] Chen X, Cheng Z, Cai H, Saunier N, Sun L: Laplacian Convolutional Representation for Traffic Time Series Imputation. TKDE 2024
>
> Thank you for your insightful comment. In response to your concern, we have examined each mentioned work. Of these 6 papers, we added [5] and [6] as new baselines. [3] is a general time series model not specifically for the time series imputation task, thus it doesn't directly fit in our codebase or field of research. [1], [2] and [4] didn't provide executable code.
>
> In summary, our model significantly outperforms the two new baselines, **with an average reduction of 79.477% and 80.247% in imputation MAE against adaTIDER [5] and LCR [6], respectively.**
>
> Moreover, we've also gathered results from the original papers of [1], [2], and [4] for a direct comparison. Some results are omitted because the corresponding original papers did not use the respective datasets.
>
> | Dataset & Mask | Score-cdm [1] | HSPGNN [2] | Motif-GNN [4] | GraphTSI (our Model) |
> | :------------: | :-----------: | :--------: | :-----------: | :------------------: |
> |     AQI36      |      --       |   11.25    |      --       |       10.3630        |
> |      AQI       |      --       |   13.30    |     15.43     |       13.2298        |
> | PEMSBAY_point  |     0.65      |    0.78    |     17.25     |        0.5692        |
> | PEMSBAY_block  |     1.55      |    1.10    |     19.04     |        0.9473        |
> |  METRLA_point  |     1.93      |     --     |      --       |        1.6339        |
> |  METRLA_block  |     2.60      |     --     |      --       |        2.2353        |
>
> As we can see in the results, **our model consistently outperforms the new baselines from [1,2,4,5,6].** We've updated the results of [2] and [4] in Table 1 of our $\\underline{\\text{revised paper}}$.
>
> > W5: It would be helpful to evaluate the model under diverse missing patterns (disjoint, MCAR, Overlap, Blackout)
>
> Thank you for your insightful comment and suggestion. The point and block missing setup in our experiments is equivalent to that of an MCAR (missing completely at random) setup. Of the four missing patterns mentioned, disjoint is a weakened missing pattern guaranteeing no two sensors experience failure at the same timestep. Therefore, we have added new experiments testing model performance under Overlap and Blackout missing patterns. Additionally, we've added a third extreme missing pattern where an entire sensor blackouts as a reference of model performance under sensor failures. The results are as follows:
>
> | Masking Pattern |  GSLI   |  Imputeformer  | GraphTSI (ours) | Improvement |
> | :-------------: | :-----: | :------------: | :-------------: | :---------: |
> |     Overlap     | 7.7746  | $\\underline{6.8932}$  |   **6.6936**    |  -2.8956%   |
> |    Blackout     | 38.1689 | $\\underline{29.1509}$ |   **13.5714**   |  -53.4443%  |
> | Sensor Failure  | 10.6389 | $\\underline{10.1796}$ |   **9.3771**    |  -7.8834%   |
>
> As we can see, GraphTSI consistently outperforms previous SOTA models under all three extreme missing patterns, further demonstrating the robustness of our proposed model. We incorporated these results in Appendix C.4 of our $\\underline{\\text{revised paper}}$ for your review.

---

### Official Review · Reviewer_PEkM · 2025-10-26

**Soundness:** 3
**Presentation:** 3
**Contribution:** 3
**Rating:** 8
**Confidence:** 4

**Summary:**

To tackle the multivariate time series imputation problem, this paper introduces GraphTSI which separates the signal and noise in time series data and utilize a bipartite graph constructed between temporal nodes and spatial nodes. By using transformer architecture and graph message passing methods, the proposed GraphTSI can outperform multiple strong baselines on multiple real-world datasets. Extensive experiments such as ablation study, missing rates analysis, as well as downstream tasks have been conducted to prove the effectiveness of the model.

**Strengths:**

1. The paper is well written with clear figures, tables, equations. The motivation and idea of the models in the paper is also grounded.
2. Extensive empirical experiments as well as theoretical analysis have been included in this paper to support the claims and demonstrate GraphTSI’s effectiveness.
3. Anonymous codes have been provided for better reproducibility of the proposed method.

**Weaknesses:**

1. Although the proposed GraphTSI shows compelling performance compared to other baselines, there is no efficiency / complexity analysis of the algorithms compared to others. It would help strengthen the contributions of GraphTSI if it is relatively efficient at the same time.
2. Theorem 1 in appendix is valid when the assumption of noise independence holds between different observations. However, it might not always hold true in real world since some noises collected in data might come from weather or human activities, etc.



**Suggestions**
1. There is a typo for section 6 name: it should be “conclusion” instead of “consluion”.
2. To help address weakness 2, it would be great if the authors can conduct some experiments by adding some manually crafted noise such as temporal dependent noise to the data. Analyzing the model performance and the signal/noise separate modeling under such cases would strengthen the contribution of the paper.

**Questions:**

1. Since transformer is able to handle extremely long window sizes, I wonder if the authors have tried much larger windows sizes like 512 / 1024. What’s the performance? Is there any bottleneck from the algorithms for instance the bipartite graph etc.? Will model having larger windows sizes with more temporal receptive field help improve the imputation?

---

> ### Author Response · Authors · 2025-11-24
> **Response to PEkM [Part 1]**
>
> We would like to sincerely thank Reviewer PEkM for your constructive review and comments. Below are our responses to your concerns and questions.
>
> ---
>
> **1. Regarding model efficiency**
>
> > W1: Although the proposed GraphTSI shows compelling performance compared to other baselines, there is no efficiency / complexity analysis of the algorithms compared to others. It would help strengthen the contributions of GraphTSI if it is relatively efficient at the same time.
>
> Thank you very much for admitting our work. In general, our model consists of three main parts: the missing-aware input embedding, the bi-unidirectional predictors, and the augmented bipartite graph. Specifically, given a windowed series of size $W\\times S\\times C$, where $W$ is the window size, $S$ is the number of series, and $C$ is the number of channels per series:
>
> - **missing-aware input embedding:** $O(WNCd\_{\\mathrm{h}})$, where $d\_{\\mathrm{h}}$ is the size of hidden states;
> - **bi-unidirectional predictors:** $O(W^2Nd\_{\\mathrm{h}})$, since attention scales quadratically with length
> - **augmented bipartite graph:** gather and diapatch of signal component is $O(WHNd\_{\\mathrm{h}})$; gather and dispatch of noise component is $O(WNd\_{\\mathrm{h}})$; propagation of signal component is $O(\\left|\\mathcal{E}^{\\mathrm{S}}\\right|Hd\_{\\mathrm{h}})$; and propagation of noise component is $O(\\left|\\mathcal{E}^{\\mathrm{S}}\\right|Nd\_{\\mathrm{h}})$, where $H$ is the number of latent heads, and $\\left|\\mathcal{E}^{\\mathrm{S}}\\right|$ is the number of edges in exgenous spatial graph.
>
> Together, the model has complexity $O\\left(WNd\_{\\mathrm{h}}\\left(W+H+C\\right)+\\left|\\mathcal{E}^{\\mathrm{S}}\\right|d\_{\\mathrm{h}}(H+N)\\right)$, which scales quadratically with window sizes.
>
> In response to your suggestions, we've included an analysis of model complexity in Section 4.4 of our $\\underline{\\text{revised paper}}$. Furthermore, in Section 5.2 of our $\\underline{\\text{revised paper}}$, we've included a new experiment regarding model efficiency against other methods. Figure 6 of our $\\underline{\\text{revised paper}}$ demonstrated model speed for inference over relative imputation error. From these observations, we can see that GraphTSI delivers lower imputation error while performing just as fast as other baseline models.
>
> We hope this additional information addresses your questions and concerns.
>
> **2. Regarding the made assumptions**
>
> > W2: Theorem 1 in appendix is valid when the assumption of noise independence holds between different observations. However, it might not always hold true in real world since some noises collected in data might come from weather or human activities, etc.
>
> > Suggestions2: To help address weakness 2, it would be great if the authors can conduct some experiments by adding some manually crafted noise such as temporal dependent noise to the data. Analyzing the model performance and the signal/noise separate modeling under such cases would strengthen the contribution of the paper.
>
> Thank you so much for your comment and suggestions. We acknowledge the fact that real-world *noise* collected in data originating from weather or human activities may exhibit temporal correlations. However, we would like to clarify that **the signal and noise are names we've attributed to the components our Data Generating Process and its assumptions create, and may differ from the general definition of these terms in what they represent.** Here, we provide a simplified proof to validate these assumptions:
>
> Suppose the noise does exhibit temporal correlation (To avoid confusion, hereafter we refer to temporally correlated noise as **shock**, and the other component as **trend**). We can further decompose it into what is predictable from past information using its autocorrelation property and the unpredictable remainder. Since predicting this shock uses only past information, we can incorporate this part with the **trend**, which we simply refer to as the **signal**, and the remainder as **noise**. Now, the noise is temporarily uncorrelated, and we've transformed the original components (**trend** and **shock**) to DGP-compliant components (**signal** and **noise**), **guaranteeing the generality of our assumptions.** We've included a more detailed mathematical proof in Appendix A.3 of our $\\underline{\\text{revised paper}}$.
>
> In response to your suggestions, we've added an additional case study in Appendix D.2 of our $\\underline{\\text{revised paper}}$, comparing the signal and noise components decomposed by the model against the theoretical value of **trend** and **shock**, as well as **signal** and **noise** using an artificially generated dataset. **The results provide empirical evidence that the model decomposes the original series into explainable components compliant with our assumptions.**
>
> We hope this additional information addresses your questions and concerns.

---

> ### Author Response · Authors · 2025-11-24
> **Response to PEkM [Part 2]**
>
> **3. Regarding performance under longer window sizes**
>
> > Q1: Since transformer is able to handle extremely long window sizes, I wonder if the authors have tried much larger windows sizes like 512 / 1024. What’s the performance? Is there any bottleneck from the algorithms for instance the bipartite graph etc.? Will model having larger windows sizes with more temporal receptive field help improve the imputation?
>
> Thank you so much for your insightful question. As the window size increases, we observe that imputation error (MAE) stabilizes and does not further decrease for larger window sizes. The results have been integrated into Figure 4\(c\) and Figure 8 of our $\\underline{\\text{revised paper}}$. To illustrate this trend, we've added more experiments regarding longer window sizes. Due to hardware and time limits, we are unable to run extremely long window sizes 512/1024 as you suggested. Instead, we've added new imputation results under window sizes 64 and 128. The results are as follows:
>
> | window size | PEMS08 MAE | Imputation Speed | METR_LA MAE | Imputation Speed |
> | :---------: | :--------: | :--------------: | :---------: | :--------------: |
> |      3      |   2.0916   |   0.03 s/iter    |   1.7551    |   0.02 s/iter    |
> |      6      |   1.8895   |   0.03 s/iter    |   1.6476    |   0.03 s/iter    |
> |     12      |   1.8357   |   0.05 s/iter    |   1.6384    |   0.05 s/iter    |
> |     24      |   1.8007   |   0.13 s/iter    |   1.6339    |   0.10 s/iter    |
> |     36      |   1.8079   |   0.15 s/iter    |   1.6433    |   0.14 s/iter    |
> |     64      |   1.8257   |   0.17 s/iter    |   1.6485    |   0.18 s/iter    |
> |     128     |   1.8573   |   0.27 s/iter    |   1.6632    |   0.18 s/iter    |
>
> **Regarding long window sizes:** The reason why the model doesn't benefit from extremely large window sizes is that imputation inherently exhibits strong locality, where nearby information plays a more critical role for imputation than distant context. To prove this, we collected the attention matrix of the forward bi-unidirectional predictor and presented the attention weight across all heads and layers, aggregated as the 99th percentile over the batch and sensor dimension in Appendix D.1 of the $\\underline{\\text{revised paper}}$. Of these attention heads, we can identify three primary patterns: 1) heads that aggregate information from the beginning of each series; 2) heads that exhibit a pronounced periodic structure that decays gradually with increasing lag; 3) the majority of heads display strong locality, focusing predominantly on a narrow temporal window preceding each time step.
>
> **Regarding model bottlenecks:** **Our bipartite graph–based algorithm DOES NOT suffer from an efficiency bottleneck**, as its complexity scales linearly with respect to the time window size. In contrast, the transformer architecture exhibits quadratic time complexity, which introduces an efficiency bottleneck for larger window sizes. However, we observed strong locality in the data, as discussed above. Therefore, it may be wise to swap out the the classical transformer within the bi-unidirectional predictor with a more efficient architecture, such as a discrete attention transformer. This substitution would enable linear scalability with respect to the window size while maintaining comparable performance. And we plan to explore these possibilities in future works.
>
> We hope this additional information addresses your question.
>
> **4. Regarding the writing of the paper**
>
> > Suggestions1: There is a typo for section 6 name: it should be “conclusion” instead of “consluion”.
>
> We express our sincerest gratitude for your helpful suggestion. We have changed this typo and carefully proofread our $\\underline{\\text{revised paper}}$ for your review.

---

### Official Review · Reviewer_F8oe · 2025-10-30

**Soundness:** 3
**Presentation:** 2
**Contribution:** 2
**Rating:** 4
**Confidence:** 4

**Summary:**

The paper introduces a graph-based method that incorporates bidirectional attention layers and spatial attention blocks within a two-step propagation scheme using virtual spatial nodes (analogous to several hierarchical attention methods proposed in the literature). Moreover, the architecture includes a mechanism to refine predictions based on the error at observed nodes. The resulting architecture achieves remarkable performance on a solid selection of datasets. Despite this, the technical novelty of the paper is overstated and relatively limited, and there are also some issues in the empirical evaluation.

**Strengths:**

* Strong empirical performance.
* The architecture design appears sound and well-motivated.
* Using reconstruction error as input to refine imputations is an interesting idea, though not novel.

**Weaknesses:**

### Main comments

* **Overstated technical novelty.**  The claimed technical novelty is overstated and relatively limited.
  - The paper presents as its main contribution the introduction of a component that propagates spatiotemporal information using a bipartite graph. However, this approach is equivalent to **many** hierarchical attention mechanisms that rely on hub nodes and two-step propagation (see, e.g., [1, 2] and hierarchical attention in [3]). Similar ideas have also been applied in spatiotemporal graph-based models (e.g., [4]).
  - The idea of feeding prediction residuals back into the model to refine predictions is reasonable but well-established. It is a known technique in time series analysis — for instance, in ARIMA models and, more recently, in neural architectures such as [5].
  - The paper claims that existing architectures struggle to propagate information effectively. However, as mentioned above, the literature has already explored a wide variety of mechanisms for effective spatiotemporal propagation. For example, the abstract states that prior models “suffer from two key limitations: (1) treating each time series as an indivisible whole, without uncovering its internal temporal dynamics, and (2) relying on linear projections to connect spatial and temporal representations.” This is hardly true for many modern architectures (e.g., [6]) that integrate spatial and temporal propagation more effectively.

* **Possible issue in the empirical evaluation.**  In Appendix A.2, the authors mention training on simulated missing data in the training set. However, this setup would be unrealistic in the real-world scenarios that those missing data are simulating (since one cannot train on data that are actually missing). Many of the included baselines (e.g., [7]) assume that simulated missing data are *not* available during training. Could the authors clarify this point? Are all baselines trained on the same ground-truth data?

Given these issues and limitations, I cannot recommend acceptance at this stage.

### Minor comments / suggestions

* One component of the model is described as modeling the *noise* part of the signal. However, this can be misleading. Claiming that the model “extracts” the noise component implies assuming the predictor is optimal and fully captures all temporal dependencies. It might be more appropriate to describe this as modeling the **error** or **residual** from the first imputation stage, which is then refined. Moreover, stating that correlations in the noise component are purely spatial seems questionable: for example, the “system-wide incidents” mentioned would likely persist for more than one timestep.
* In the propagation step (line 320), how are the neighbors of each node defined? Is the graph fully connected? If so, why refer to it as a graph — how does it differ from standard spatial attention?
* Many cited papers refer to their arxiv versions despite having published counterparts; please update the references accordingly.

---

**References**


[1] Ravula et al., "ETC: Encoding long and structured inputs in Transformers", EMNLP 2020\\

[2] Dolga et al., "Latte: Latent Attention for Linear Time Transformers" arxiv 2024\\

[3] Marisca et al., "Learning to Reconstruct Missing Data from Spatiotemporal Graphs with Sparse Observations", NeurIPS 2022 \\

[4] Satorras et al., "Multivariate time series forecasting with latent graph inference", arxiv 2022\\

[5] Kim et al., "Residual Correction in Real-Time Traffic Forecasting", CIKM 2022\\

[6] Wu et al., "Traversenet: Unifying space and time in message passing for traffic forecasting" TNNLS 2022\\

[7] Cini et al., "Filling the G_ap_s: Multivariate Time Series Imputation by Graph Neural Networks", ICLR 2022

**Questions:**

Please comment on the above weaknesses.

---

> ### Author Response · Authors · 2025-11-24
> **Response to Reviewer F8oe [Part 1]**
>
> Thank you for your insightful comments. We have addressed your specific concerns and questions as detailed below:
>
> ---
>
> **1. Regarding our technical novelty**
>
> > The paper presents as its main contribution the introduction of a component that propagates spatiotemporal information using a bipartite graph. However, this approach is equivalent to many hierarchical attention mechanisms that rely on hub nodes and two-step propagation (see, e.g., [1, 2] and hierarchical attention in [3]). Similar ideas have also been applied in spatiotemporal graph-based models (e.g., [4]).
> >
> > [1] Ravula et al., "ETC: Encoding long and structured inputs in Transformers", EMNLP 2020
> >
> > [2] Dolga et al., "Latte: Latent Attention for Linear Time Transformers" arxiv 2024
> >
> > [3] Marisca et al., "Learning to Reconstruct Missing Data from Spatiotemporal Graphs with Sparse Observations", NeurIPS 2022
> >
> > [4] Satorras et al., "Multivariate time series forecasting with latent graph inference", arxiv 2022
>
> Thank you for your detailed comments. In our augmented bipartite graph component, **the spatial exogenous graph defined as $\\mathcal{G}^{\\mathrm{S}}$ is responsible for the propagation of spatial information among sensors**, whereas the **bipartite subgraph** $\\mathcal{G}^{\\mathrm{bip}}$ is responsible for the transformation between spatial and temporal representations. **Together, they form the augmented bipartite graph**, which has additional edges (spatial exogenous graph edges) on the spatial representation side of the bipartite graph.
>
> While we acknowledge the resemblance to [1,2,3,4] in employing an auxiliary node set for information transmission, our approach differs substantially in its specific implementation and methodological design in three major ways:
>
> 1. **The representation semantics**: The attention mechanism in these mentioned works forces a homogenous representation onto both sides of the node set, whereas our design adopts non-linear transformation functions to adapt between spatial and temporal representation, enabling each side to prioritize distinct semantic cues.
> 2. **Design target**: Most of the mentioned work introduces an auxiliary node set to reduce the complexity of performing attention or self-attention, whereas our design focuses on adaptively transforming spatial representation with temporal representation for different time series components.
> 3. **Information propagation among auxiliary nodes**: Most of the mentioned work uses the attention mechanism to exchange information among auxiliary nodes, whereas our design propagates information via the spatial exogenous graph with fixed learned edge weights during inference, effectively capturing stationary dependencies.
>
> To elaborate, we compare our design against each of the mentioned works:
>
> **Against ETC [1]**
>
> - **Homogeneous vs. Heterogeneous Information**: ETC (and other hierarchical attention) collects global information on auxiliary node sets; both global and local nodes **share the same representation** (i.e., they lie in the same semantic and feature space). By contrast, our augmented bipartite graph **exhibits different meanings** on its two node sets: spatial representation and temporal representation, and are interconverted via artificially designed gather and dispatch functions.
> - **Information Propagation**: ETC shares global information using **dynamically computed attention matrices**, whereas our augmented bipartite graph propagates information using learned graph edge weights that **remain fixed at inference**, which is critical to capture stationary spatial relationships.
>
> **Against Latte [2]**
>
> - Latte aggregates latent global information on its auxiliary node sets, and is specifically designed to accelerate large-model training via two-stage attention mechanisms for linear complexity transformers. **There is no information exchange among the latent tokens in Latte**.
> - Latte's attention mechanisms are optimized approximations of the standard attention for **homogeneous representations**. In contrast, our augmented bipartite graph employs dedicated gathering and dispatch functions to transform between spatial and temporal representations, enabling each side to prioritize distinct semantic cues.
>
> **Against SPIN [3]**
>
> - To aggregate information from related sensors, SPIN uses spatial-temporal cross-attention, which directly queries all non-missing observations from related sensors, and leverages auxiliary node sets as latent global information to speed up attention. **Similar to ETC, this setup requires the auxiliary nodes to share the same representation space as the original node set to compute attention.**
> - Similar to Latte, **there is no information exchange among the global latent representations.**
>
> **[Continued in Part 2...]**

---

> ### Author Response · Authors · 2025-11-24
> **Response to F8oe [Part 2]**
>
> **[Continued from Part 1]**
>
> **Against FC-GNN and BP-GNN [4]**
>
> - Both FC-GNN and BP-GNN **directly map** entire univariate time series to spatial representation without modeling intra-series temporal interactions. In contrast, our model first constructs semantically rich temporal representations before the augmented bipartite graph transforms them to spatial representations.
> - Both FC-GNN and BP-GNN **use an MLP for transforming raw timeseries into spatial representations**, whereas our design leverages **distinct gather and dispatch functions to capture component-specific features and improve separation**.
>
> > The idea of feeding prediction residuals back into the model to refine predictions is reasonable but well-established. It is a known technique in time series analysis — for instance, in ARIMA models and, more recently, in neural architectures such as [5].
> >
> > [5] Kim et al., "Residual Correction in Real-Time Traffic Forecasting", CIKM 2022
>
> Thank you for your insightful comment. However, we would like to clarify that our proposed subtraction-prediction framework is **significantly different from** the residually connected prediction refinement. We've listed a few primary differences as follows:
>
> - **Design Differences**. Our prediction-subtraction framework progressively estimates different time series components (i.e., signal and noise) and eventually joins them together to form the final imputation. Compared to the residually connected imputation methods, which iteratively refine prediction to form the final imputation, our design primarily differs in two ways: 1) the **residual connection repeats the same model structure** multiple times, whereas **our framework uses *different* custom-designed gather and dispatch functions for the signal and noise components**; 2) **residual connection create unexplainable gradual refinements**, whereas **our framework creates explainable signal-noise components** that may well scale into more components according to the specific given knowledge of each time series. We've **added additional experiments in Appendix D.2** of our $\\underline{\\text{revised paper}}$ to further validate the merits of our explainability.
> - **Usage Differences: The residual-based methodology, which emphasizes inter-layer iterative refinement, is orthogonal to our proposed framework that focuses on intra-layer decomposition**. Feeding prediction residual back into the model is equivalent to adding more layers to our model, where each layer iteratively refines our imputation. **The prediction-subtraction framework is a separate component inside each layer of the model.** We have not made any claims regarding the performance improvement through the residually-connected multi-layer setup. Instead, our focus lies in introducing an extensible and explainable framework that empirically improves performance through decomposing time series into multiple components and handling each separately.
> - We acknowledge that the mentioned work [5] did establish the fact that forecasted residuals exhibit autocorrelation. However, **their model is primarily an additive component to existing forecasting backbone models, and is far from being well-established in the processing and utilization of such residual components**. Furthermore, the forecasting and imputation domains, while related, entail significant methodological differences: imputation involves reconstructing missing values from observed fragments using bi-directional context, whereas forecasting relies on fully observed histories for uni-directional extrapolation.

---

> ### Author Response · Authors · 2025-11-24
> **Response to Reviewer F8oe [Part 3]**
>
> > The paper claims that existing architectures struggle to propagate information effectively. However, as mentioned above, the literature has already explored a wide variety of mechanisms for effective spatiotemporal propagation. For example, the abstract states that prior models “suffer from two key limitations: (1) treating each time series as an indivisible whole, without uncovering its internal temporal dynamics, and (2) relying on linear projections to connect spatial and temporal representations.” This is hardly true for many modern architectures (e.g., [6]) that integrate spatial and temporal propagation more effectively.
> >
> > [6] Wu et al., "Traversenet: Unifying space and time in message passing for traffic forecasting" TNNLS 2022
>
> Thank you for your insightful comments. We would like to respectfully clarify that:
>
> - **Our argument is not that these models fail to integrate spatiotemporal information, but that how this spatiotemporal information is leveraged can be further refined.** Specifically, while models like [6] or [3] effectively utilize information from a sensor's own history as well as other sensors' past observations, **the use of direct attention mechanism forces each node to share the same spatiotemporal representation within a unified embedding space**. We posit that this "shared representation" paradigm may **not be optimal for capturing the inherently distinct pattern between spatial dependency and temporal dynamics**. In contrast, our model separately introduces the bi-unidirectional predictor to capture temporal dynamics, and the spatial exogenous graph within the augmented bipartite graph to capture spatial dependencies, and connects the two parts via the bipartite subgraph. **This enables each component to prioritize distinct semantic cues of spatial or temproal interactions.**
> - Furthermore, we would like to note that our work focuses on the multivariate imputation task, which presents different challenges and data characteristics compared to the forecasting tasks addressed in [6]. We propose a whole set of techniques for reconstructing missing values from observed fragments using bi-directional context, while forecasting relies on fully observed histories for uni-directional extrapolation.
>
> In the $\\underline{\\text{revised paper}}$, we have added more discussions regarding the limitations of [3,4,5,6], and highlight the differenes of our model.
>
> **2. Regarding details of empirical evaluation**
>
> > In Appendix A.2, the authors mention training on simulated missing data in the training set. However, this setup would be unrealistic in the real-world scenarios that those missing data are simulating (since one cannot train on data that are actually missing). Many of the included baselines (e.g., [7]) assume that simulated missing data are *not* available during training. Could the authors clarify this point? Are all baselines trained on the same ground-truth data?
> >
> > [7] Cini et al., "Filling the G_ap_s: Multivariate Time Series Imputation by Graph Neural Networks", ICLR 2022
>
> Thank you for your question. **Yes, all baselines are trained on the same ground truth data with identical masks. And, yes, we've referenced many previous works and codes, including [7], when testing our model.** We have adopted the same training setup described in [7] (as indicated by both the manuscript and open-source code), which both train on simulated missing data. [7] and our setting both guarantee that **no information** (both the simulated missing ones and the non-missing ones) from the validation or testing set will ever be shown to the model during training.
>
> However, to ease your concern regarding the practicality of this setup, we've tested both **AQI** and **AQI36** datasets, which lie closely to the proposed real-world scenarios. For these two datasets, roughly 25% of their original observations are irrecoverably lost. To train our model on these datasets, we follow previous works [8] and artificially drop an observation for training and evaluation if 1) the observation is not missing, and 2) if the observation from the same time last month is missing. **This creates simulated missing masks that follow natural missing patterns, and allows us to train the model without knowing the actual missing observations.** Our model achieved 10.383% and 17.678% improvements in MAE over the best-performing baseline on AQI and AQI36, respectively. Demonstrating the superior imputation ability in datasets with real-world missing patterns.
>
> **References:**
>
> - [8] Xiuwen Yi, Yu Zheng, Junbo Zhang, and Tianrui Li. St-mvl: Filling missing values in geo-sensory time series data. In Proceedings of the 25th international joint conference on artificial intelligence, 2016.

---

> ### Author Response · Authors · 2025-11-24
> **Response to Reviewer F8oe [Part 4]**
>
> **3. Regarding implementation detail**
>
> > One component of the model is described as modeling the *noise* part of the signal. However, this can be misleading. Claiming that the model “extracts” the noise component implies assuming the predictor is optimal and fully captures all temporal dependencies. It might be more appropriate to describe this as modeling the **error** or **residual** from the first imputation stage, which is then refined. Moreover, stating that correlations in the noise component are purely spatial seems questionable: for example, the “system-wide incidents” mentioned would likely persist for more than one timestep.
>
> Thank you for your insightful question. As we've stated before, our main contribution in this paper is that we design a prediction-subtraction mechanism within each layer to artificially decompose time series into various temporal components, rather than performing a residually connected refinement. We use "*extract*" to describe the explicit architectural separation of the learned signal from the remaining components. We do not claim the predictor is optimal; rather, this separation allows for better explainability of the signal and noise.
>
> Regarding your questions on our claim that the noise component only exhibits spatial correlation, we would like to respectfully clarify that **both signal and noise are defined via our proposed Data Generating Process and its assumptions, and what the signal and noise represent under these assumptions may differ from the general definition of these terms.** Here, we provide a simplified proof to validate these assumptions:
>
> Suppose the noise does exhibit temporal correlation (To avoid confusion, hereafter we refer to temporally correlated noise as **shock**, and the other component as **trend**). We can further decompose it into what is predictable from past information using its autocorrelation property and the unpredictable remainder. Since predicting this shock uses only past information, we can incorporate this part with the **trend**, which we simply refer to as the **signal**, and the remainder as **noise**. Now, the noise is temporarily uncorrelated, and we've transformed the original components (**trend** and **shock**) to DGP-compliant components (**signal** and **noise**), **guaranteeing the generality of our assumptions.**
>
> We've included a more detailed mathematical proof in Appendix A.3 of our $\\underline{\\text{revised paper}}$. And in Appendix D.2, we've added an additional experiment comparing the signal and noise components decomposed by the model against the theoretical value of **trend** and **shock**, as well as **signal** and **noise** using an artificially generated dataset. **The results provide empirical evidence that the model decomposes the original series into explainable components compliant with our assumptions.**
>
> We hope this additional information addresses your questions.
>
> > In the propagation step (line 320), how are the neighbors of each node defined? Is the graph fully connected? If so, why refer to it as a graph — how does it differ from standard spatial attention?
>
> Thank you for your question. The neighbours during the propagation mechanism $j\\in\\mathrm{N}\\left(i,\\mathcal{E}^\\mathrm{S}\\right)$ are defined as all spatial nodes $j\\in\\mathcal{V}^\\mathrm{S}$ directly connected to node $i$ via the spatial exogenous graph $\\mathcal{G}^\\mathrm{S}$, which is given in the dataset to depict the spatial relationships among sensors.
>
> In the AQI and AQI36 datasets, this would be the clamped exponentially decayed geographical distance; In the traffic datasets, this would be the actual roadmaps. We only use a fully-connected graph as a fallback for datasets without predefined graph information. Also, we only use the graph structure and learn edge weights during training to capture more versatile relationships.
>
> The propagation mechanism alone is similar to a message-passing process with learnable edge weights. Attention calculates attention weights according to hidden state vectors, whereas our propagation implementation uses edge weights that are learnable during training but remain fixed during inference regardless of the hidden vector values. We chose this design as it resembles more to the real world, where relationships between sensors are generally fixed and should not constantly change during operation.
>
> We hope this additional information addresses your questions. We've also updated related sections in our $\\underline{\\text{revised paper}}$ to better explain how neighbours and information exchange processes are defined.
>
> **4. Regarding the writing of the paper**
>
> > Many cited papers refer to their arxiv versions despite having published counterparts; please update the references accordingly.
>
> We would like to sincerely thank you for your helpful comment. In response to your comment, we've updated all the references accordingly in our $\\underline{\\text{revised paper}}$.

---

### Official Review · Reviewer_cqdy · 2025-11-01

**Soundness:** 1
**Presentation:** 1
**Contribution:** 1
**Rating:** 2
**Confidence:** 4

**Summary:**

This paper proposes GraphTSI, a graph-based framework for multivariate time series imputation that integrates signal–noise decomposition with spatial–temporal graph propagation. The method introduces a prediction–subtraction mechanism to separate predictable signal components from unpredictable noise and employs an augmented bipartite graph to model non-linear spatial–temporal interactions. Experiments on nine real-world datasets reportedly show that GraphTSI outperforms recent graph-based and transformer-based methods in imputation accuracy.

**Strengths:**

- The idea of combining signal–noise decomposition with graph-based spatial–temporal modeling is conceptually interesting and may help disentangle predictable and unpredictable components in temporal systems.

- The proposed prediction–subtraction framework and augmented bipartite graph demonstrate an effort to move beyond standard linear projections.

- The experiments cover multiple real-world datasets with ablation and robustness analyses, showing reasonable empirical effort.

**Weaknesses:**

Overall, the presentation requires substantial improvement before the technical contributions can be properly evaluated. Key weaknesses include:

- **Weak motivation**: Each proposed module (signal–noise decomposition, prediction–subtraction, augmented bipartite graph) lacks a clear rationale for its necessity or effectiveness.

- **Limited methodological depth**: The paper describes design choices without strong theoretical or empirical justification, making it difficult to assess novelty or soundness.

- **Poor writing quality**: The exposition is hard to follow, with unclear transitions and dense descriptions that obscure the main ideas.

- **Others**: The text in Figures 1 and 2 is too small to read comfortably, further hindering understanding.

**Questions:**

1. The paper claims that standard MLPs cannot distinguish missing from observed values and introduces a miss-aware embedding. However, Equation (5) does not clearly show how missingness is encoded. Does it explicitly use the binary mask, or rely on separate learnable parameters? A clearer mathematical explanation or ablation is needed.
2.  In *Augmented Bipartite Graph Transformation,* the definitions of $\mathcal{V}^{\mathrm{T}}$ and $\mathcal{V}^{\mathrm{S}}$ are unclear. How are these node embeddings initialized or derived—via separate encoders or direct parameterization? Clarification would improve reproducibility and understanding.

---

> ### Author Response · Authors · 2025-11-24
> **Response to reviewer cqdy [Part 1]**
>
> > Weak motivation: Each proposed module (signal–noise decomposition, prediction–subtraction, augmented bipartite graph) lacks a clear rationale for its necessity or effectiveness.
>
> We thank the reviewer for the feedback. However, we would like to clarify that our modules are both necessary and coherently connected, with the overall design informed by the proposed data-generating process (DGP). The DGP specifies a decomposition of each series into signal and noise, which in turn dictates the following components.
>
> - **A bi-unidirectional predictor** that, at each pass, uses only past or only future information to capture univariate predictable signals without temporal leakage.
> - **An augmented bipartite graph** that exchanges the resulting univariate signal, as well as the decomposed univariate noise estimates across sensors through the exogenous spatial graph, using dedicated gather and dispatch functions to exploit temporal redundancy of signal components, and different gather and dispatch functions to preserve sudden, non-repetitive characteristics of noise components.
> - **A prediction–subtraction module** that extracts noise estimates from the original non-missing observations according to our predicted signal components.
>
> This sequence: signal prediction, cross-node sharing, residual extraction, and residual-level exchange, follows directly from the DGP and is required to address the distinct yet complementary sub-problems in multivariate imputation.
>
> We evaluated GraphTSI on 12 datasets across 4  domains against 9 competitive baselines and found that it consistently outperforms all methods in imputation MAE (see Table 1 of our $\\underline{\\text{revised paper}}$), with average improvements of 17.58% and 10.27% over the prior SOTA models Imputeformer and GSLI, respectively. We further assess: 1) the effectiveness of each module via **ablations**; 2) imputation quality via downstream classification; 3) robustness under **extremely high missing rates** up to 95%; 4) training stability under **varying hyperparameters**; and 5) model **efficiency**. These results and analysis are incorporated in Section 5.2 of our $\\underline{\\text{revised paper}}$.
>
> Additional results on **missing rates, long window sizes, and different missing patterns** are provided in Appendix C of the $\\underline{\\text{revised paper}}$. In addition, Appendix D of our $\\underline{\\text{revised paper}}$ includes case studies of **attention matrices** and **visualizations of decomposed signal and noise components** on artificially generated datasets, supporting the model’s **rationale and explainability.**
>
> In summary, **we articulate a clear design rationale for GraphTSI and demonstrate its robustness and effectiveness through extensive experiments**.
>
> > Limited methodological depth: The paper describes design choices without strong theoretical or empirical justification, making it difficult to assess novelty or soundness.
>
> We would like to respectfully clarify that the paper provides a solid mathematical foundation for our design choices. In Section 2 of our $\\underline{\\text{revised paper}}$, we: 1) **define the preliminaries** of multivariate time-series imputation and 2) **present our modeling of the data-generating process (DGP)**, which informs the overall model design. The validity of our assumptions is discussed in Appendix A.3 of our $\\underline{\\text{revised paper}}$, and we include two additional mathematical results in Appendix A.1 and A.2 of our $\\underline{\\text{revised paper}}$, which **establish the stability of the training procedure**, and **characterize the limitations of prior models in their spatio–temporal interconversion design**. Collectively, these results provide theoretical support for the soundness of our approach and theoretical justification for its novelty.
>
> Beyond theory, we also report empirical evidence and case studies in Appendix D of our $\\underline{\\text{revised paper}}$. Temporal **attention matrices for long window sizes** (Appendix D.1 of our $\\underline{\\text{revised paper}}$) offer **empirical support for the explainability of our design**, while **visualizations of the decomposed signal and noise components** on artifical datasets (Appendix D.2 of our $\\underline{\\text{revised paper}}$) further guarantees **the soundness of the proposed mechanism**.
>
> Additional experiments including **hyperparameter analyses, evaluations under extremely high missing rates, studies of diverse missing patterns, and ablation studies**, provide further evidence that **our model design is sound and robust across a broad range of real-world scenarios**.
>
> In summary, we have provided **both rigorous theoretical guarantees and extensive empirical validation for the design of GraphTSI**.

---

> ### Author Response · Authors · 2025-11-24
> **Response to reviewer cqdy [Part 2]**
>
> > Poor writing quality: The exposition is hard to follow, with unclear transitions and dense descriptions that obscure the main ideas.
>
> In response to your comment, we have updated the main text in places where it may cause confusion and misunderstandings. We've also updated our Appendix and organized it in a more structured hierarchy in our $\\underline{\\text{revised paper}}$ for your review.
>
> > Others: The text in Figures 1 and 2 is too small to read comfortably, further hindering understanding.
>
> Thank you for your suggestions. In the $\\underline{\\text{revised paper}}$, we have updated Figure 1 and Figure 2 with larger fontsizes. We believe these changes improve the visual quality and interpretability of the figures.
>
> > The paper claims that standard MLPs cannot distinguish missing from observed values and introduces a miss-aware embedding. However, Equation (5) does not clearly show how missingness is encoded. Does it explicitly use the binary mask, or rely on separate learnable parameters? A clearer mathematical explanation or ablation is needed.
>
> We appreciate your comment. Equation 5 is as follows:
> $$
> \\mathbf{h}\_{i,\\tau}^{(0)}=\\mathrm{MLP}\\left(\\widetilde{\\mathbf{X}}^{(t)}\_{i,\\tau}\\middle\\Vert\\mathbf{M}^{(t)}\_{i,\\tau}\\middle\\Vert\\mathbf{E}^{\\mathrm{T},(t)}\_{i,\\tau}\\right)
> $$
> where $\\widetilde{\\mathbf{X}}^{(t)}\_{i,\\tau}$ represents the incomplete multivariate time series; $\\mathbf{M}^{(t)}\_{i,\\tau}$ represents the binary missing mask; and $\\mathbf{E}^{\\mathrm{T},(t)}\_{i,\\tau}$ represents the temporal covariates, where we follow previous works [1,2] and employ a sinusoidal time-of-day encoding for each observation. No additional learnable parameters are used except for the MLP embedder. Furthermore, we would like to clarify that we didn't claim this missing-aware input embedding as our primary contribution. In fact, we clearly stated that this module was adapted from well-established previous works such as [3] which has already been well-tested in previous works.
>
> We hope this additional information may address your concerns.
>
> **References:**
>
> - [1] Ashish Vaswani, Noam Shazeer, Niki Parmar, Jakob Uszkoreit, Llion Jones, Aidan N Gomez, Łukasz Kaiser, and Illia Polosukhin. Attention is all you need. Advances in neural information processing systems, 30, 2017
> - [2] Tong Nie, Guoyang Qin, Wei Ma, Yuewen Mei, and Jian Sun. Imputeformer: Low rankness-induced transformers for generalizable spatiotemporal imputation. In Proceedings of the 30th ACM SIGKDD Conference on Knowledge Discovery and Data Mining, KDD ’24, pp. 2260–2271, New York, NY, USA, 2024. Association for Computing Machinery. ISBN 9798400704901. doi: 10.1145/3637528.3671751. URL https://doi.org/10.1145/3637528.3671751.
> - [3] Zachary C Lipton, David Kale, and Randall Wetzel. Directly modeling missing data in sequences with rnns: Improved classification of clinical time series. In Machine learning for healthcare conference, pp. 253–270. PMLR, 2016.
>
> > In Augmented Bipartite Graph Transformation, the definitions of $\\mathcal{V}^{\\mathrm{T}}$ and $\\mathcal{V}^{\\mathrm{S}}$ are unclear. How are these node embeddings initialized or derived—via separate encoders or direct parameterization? Clarification would improve reproducibility and understanding.
>
> Thank you for your comment. In the augmented bipartite graph transformation, the $\\mathcal{V}^{\\mathrm{T}}$ and $\\mathcal{V}^{\\mathrm{S}}$ are defined as $\\mathcal{V}^{\\mathrm{T}}=\\left\\{v^t\_1,\\dots,v^t\_W\\right\\}$, and $\\mathcal{V}^{\\mathrm{S}}=\\left\\{v^s\_1,\\dots,v^s\_N\\right\\}$, which were clearly stated in lines 297 and 298 of the $\\underline{\\text{original paper}}$, and also in line 319 of our $\\underline{\\text{revised paper}}$. The actual nodes $v\_{\\cdot}^t$ and $v\_{\\cdot}^s$ have no actual node embeddings and simply act as mathematical references to better describe the structure of our augmented bipartite graph. The information that propagates on the augmented bipartite graph was stated in the following Sections 4.3.1 and 4.3.2 of both the $\\underline{\\text{original paper}}$ and the $\\underline{\\text{revised paper}}$. Since there are no node embeddings for these structure references, there is no need to separate encoders or parameterization techniques.
>
> We hope these additional information may address your questions and concerns.

---

### Author Response · Authors · 2025-12-02
**Brief summary of reviews, responses and revisions**

We sincerely thank all reviewers for their insightful and constructive comments, which have greatly helped us improve our paper. We are encouraged by the overall positive opinion of our work, highlighting:

- **Comprehensive and extensive empirical experiments** (cdqy, PEkM, 6ZuJ), and **strong empirical performance**. (F8oe, PEkM, 6ZuJ)
- **Sound and grounded motivations** (F8oe, PEkM, 6ZuJ)
- **Well-structured and clear paper, with clear figures, tables, and equations** (PEkM, 6ZuJ)

Furthermore, **two reviewers (F8oe, 6ZuJ) indicated they would reconsider their scores**. We have carefully addressed all questions and suggestions and updated the revised paper accordingly. The major revisions are:

**1. Model efficiency** (PEkM, 6ZuJ)

**Our model scales quadratically with window sizes** due to the use of standard attention. Additional results regarding inference time (Figure 6) show **competitive model performance while being relatively efficient**. In-depth analysis of attention heads (Figure 9) discussed **simple replacement with discrete attention decreases overall complexity to linearly with window sizes.**

**Revisions:**

- Section 4.4: Added detailed analysis on the time complexity of each component and the full model.
- Section 5.2, Figure 6: Added additional results on average inference time and relative imputation error compared against other baseline models.

**2. Data Generating Process Assumptions** (F8oe, PEkM, 6ZuJ)

Our assumptions uniquely separate signal and noise components from the mixed raw timeseries components, and **hold even for the auto-correlated exogenous shock in real-world** (Section A.3). We added **additional experiments** (Section D.2, Figure 10, 11) to demonstrate how GraphTSI behaves under an artificial auto-correlated shock, which **fits with our assumptions** that the decomposed components will not exhibit auto-correlation.

**Revisions:**

- Section A.3: Explained and mathematically proved the validity of our DGP assumptions, including the case of autocorrelated exogenous shocks.

- Section D.2, Figure 10, 11: Added additional results on decomposed components in artificial datasets, with detailed experiment setup, expected results, and actual observed results.

**3. Implementation and evaluation detail** (cqdy, F8oe, PEkM, 6ZuJ)

We **strictly followed previous works** for evaluating GraphTSI. All datasets were preprocessed with a **fixed random seed (seed=2)** before model loading, ensuring **identical missing masks and train/valid/test splits** across all compared methods. We have also clarified the operational details of key components.

**Revisions:**

- Section B.2: Added additional information on dataset setup, evaluation steps, and other implementation details.
- Section 4.3.1: Clarified on the Propagate step of our augmented bipartite graph.

**4. Additional experiments** (F8oe, PEkM, 6ZuJ)

Beyond original extensive experiments (benchmark, downstream tasks, varying missing rates, hyperparameter, case studies), we have added: 1) **More benchmark datasets**: Added ETTm1, ETTm2, and Elergone. Bringing to total to 12 datasets across 4 domains (air quality, healthcare, traffic, smart grid); 2) **Varying missing patterns**: Evaluated performance under Overlap, Blackout, and Sensor Failure scenarios. Our model consistently outperforms baselines; 3) **Long window sizes**: Evaluated performance up to $W=128$. GraphTSI maintained superior performance. Further analysis confirms the locality of attention in imputation tasks, highlighting potential for linear complexity to window size with sparse attention.

**Revisions:**

- Section 5.2, Table 1; Section C.1, Table 3: Added results on added datasets.
- Section C.4, Table 4: Added results of MAE under different missing patterns.
- Section C.3, Figure 4, 8: Updated and discussed model performance under long window sizes.
- Section D.1: Added 99th percentile attention weights of temporal predictors, and discussed the observed locality of attention.

**5. Novelty and differentiation** (cqdy, F8oe)

GraphTSI significantly differs from prior methods that use multi-layer residuals for imputation. We introduce **novel components** that separate and individually impute explainable components **within each layer**, addressing: **1) effective decomposition with missing values (Bi-unidirectional predictor), and 2) effective spatial-temporal representation conversion and propagation (Augmented bipartite graph).** We have clarified these differences from the mentioned works.

**Revisions:**

- Section 1: Added discussion of limitations that previous models force a shared spatial-temporal feature space.
- Section 3: Added discussion on the major difference of GraphTSI against previous models, and mentioned and discussed the FG-GNN models.

We believe these revisions thoroughly address the reviewers' concerns. We thank the reviewers again for their invaluable feedback, which has significantly strengthened the paper.

---

### Meta-Review · Area_Chair_RUPc · 2026-01-07

**Summary:**

This paper proposes a prediction–subtraction framework combined with an augmented bipartite graph, representing an attempt to move beyond standard linear projection approaches. The experimental evaluation covers multiple real-world datasets and includes ablation and robustness analyses, indicating a reasonable level of empirical effort. Nevertheless, despite the rebuttal, several important concerns remain insufficiently addressed. Taken together, these issues lead me to recommend rejection.
First, the motivation for combining signal–noise decomposition, prediction–subtraction, and the augmented bipartite graph is not convincingly established. While the authors provide additional explanations during the rebuttal, many of these arguments lack novelty and do not clearly justify why these components should be assembled in this particular manner (Reviewer cqdy). Moreover, the claimed technical novelty appears overstated. Several key components, as well as some of the limitations highlighted by the paper, have already been explored in prior work. Although the authors discuss differences between the proposed method and existing approaches, the essential distinctions remain limited. In particular, the two key limitations emphasized in the paper do not generally hold for many modern architectures that already integrate spatial and temporal propagation more effectively (Reviewer F8oe).
Second, the exposition of the paper is difficult to follow. Transitions between sections are unclear, and the presentation is dense, which obscures the core ideas. Some mathematical symbols are not explained at their first appearance, and understanding the proposed model often requires frequent reference to cited works, which substantially degrades the readability of the paper (Reviewer cqdy).
Finally, the paper places strong emphasis on the notions of “signal” and “noise.” In the rebuttal, the authors clarify that these terms are labels assigned to components arising from their assumed data-generating process and may differ from standard definitions. Specifically, they clarify that’ the signal and noise are names we've attributed to the components our Data Generating Process and its assumptions create, and may differ from the general definition of these terms in what they represent and they state that  past information, we can incorporate this part with the trend, which we simply refer to as the signal, and the remainder as noise.’ . This usage can be misleading for readers. Furthermore, the theoretical analysis—particularly Theorem 2—assumes that the noise component is independent of all previous observations, an assumption that is unlikely to hold in realistic settings (Reviewer F8oe, 6ZuJ, PEkM).

**Reviewer Concerns:**

**Reviewer cqdy:**

**Concerns and Rebuttal Evaluation:**

•	Weak motivation: Partly addressed. The rebuttal provides additional explanations, but the motivation for the proposed framework remains insufficient to support a clearly innovative contribution.

•	Limited methodological depth: Not adequately addressed. Although some components are clarified, many core elements closely follow prior work, and the overall methodological novelty remains limited. In addition, the theoretical analysis relies on strong assumptions (e.g., independent noise), which weaken the depth and generality of the contribution.

•	Poor writing quality: Partly addressed. Some clarifications are offered in the rebuttal, but the exposition remains difficult to follow, with unclear transitions and dense descriptions that obscure the main ideas.

•	Other concerns: Addressed.

Questions:
•	Q1: Addressed.

•	Q2: Addressed.


**Reviewer F8oe:**

**Concerns and Rebuttal Evaluation:**

•	Overstated technical novelty: Partly addressed. The rebuttal clarifies certain aspects of the proposed bipartite graph structure, which may constitute a degree of novelty. However, many components and ideas are closely related to existing work.

•	Possible issue in the empirical evaluation: Partly Addressed. There remains an unresolved inconsistency regarding the objective function in Theorem 2, which appears to require access to ground-truth simulated missing data, while the authors claim that the implementation does not rely on such ground truth.

•	Modeling of the noise component: Partly addressed. The distinction between temporally correlated noise (referred to as “shock”) and other components (referred to as “trend”) remains confusing and deviates from common usage, raising concerns about clarity and interpretability.

•	Graph definition: Addressed.

•	Concerns about excessive citations: Addressed.


**Reviewer PEkM*8

**Concerns and Rebuttal Evaluation:**

•	W1 Model efficiency: Addressed.

•	W2 Signal–noise definition: Partly addressed. The authors clarify that “signal” and “noise” are labels arising from their assumed data-generating process and may differ from standard definitions. However, this clarification does not fully resolve concerns about potential reader confusion.

•	Q1 window sizes: Addressed.


**Reviewer 6ZuJ**

**Concerns and Rebuttal Evaluation:**

•	W1–W4: Addressed.

•	Q1: Addressed.

•	Q2: Addressed.

•	Q3 (Noise assumptions): Not adequately addressed. The explanation regarding the assumption that noise is not temporally correlated remains confusing and insufficiently justified, particularly in realistic data settings.

**Reviewer Scores:**

Overall, it is unlikely that the rebuttal would lead to changes in the reviewers’ original scores, as the core concerns raised in the initial reviews remain largely unresolved.

•	Reviewer cqdy: The rebuttal provides limited clarification, but the major concerns regarding weak motivation, limited methodological depth, and unclear presentation persist. A score change is therefore unlikely.

•	Reviewer F8oe: While some minor clarifications are offered, the key issues around overstated novelty and conceptual inconsistencies (e.g., Theorem 2 and noise modeling) remain. The original scores are likely to be maintained.

•	Reviewer PEkM: Several points are addressed, but remaining ambiguity in the signal–noise interpretation limits any reassessment. The reviewer’s scores are unlikely to change.

•	Reviewer 6ZuJ: Most concerns are addressed, but confusion around the noise assumptions remains. Given its relevance to theoretical soundness, a score revision is unlikely.

---

### Decision · Program_Chairs · 2026-01-26

Reject